# CP-SLAM: Collaborative Neural Point-based SLAM

**Jiarui Hu[1], Mao Mao[1], Hujun Bao[1], Guofeng Zhang[1], Zhaopeng Cui[1]***

[1]State Key Lab of CAD&CG, Zhejiang University

## Abstract

This paper presents a collaborative implicit neural simultaneous localization and mapping (SLAM) system with RGB-D image sequences, which consists of complete front-end and back-end modules including odometry, loop detection, sub-map fusion, and global refinement. In order to enable all these modules in a unified framework, we propose a novel neural point based 3D scene representation in which each point maintains a learnable neural feature for scene encoding and is associated with a certain keyframe. Moreover, a distributed-to-centralized learning strategy is proposed for the collaborative implicit SLAM to improve consistency and cooperation. A novel global optimization framework is also proposed to improve the system accuracy like traditional bundle adjustment. Experiments on various datasets demonstrate the superiority of the proposed method in both camera tracking and mapping.

## 1   Introduction

Dense visual Simultaneous Localization and Mapping (SLAM) is an enduring fundamental challenge in computer vision, which aims to achieve 3D perception and exploration of unknown environments by self-localization and scene mapping with wide downstream applications in autonomous driving, unmanned aerial vehicle(UAV) navigation, and virtual/augmented reality (VR/AR).

The traditional visual SLAM has witnessed continuous development, leading to accurate tracking and mapping in various scenes. Most of the current prominent visual SLAM systems pay their primary attention to real-time tracking performance [29, 31, 4], while dense map reconstruction is normally achieved with the temporal fusion of additional depth input or estimated depth images, thus sensitive to noises and outliers. Some collaborative visual SLAM systems [34, 20] have also been proposed as a straightforward extension of monocular visual SLAM.

The learning-based visual SLAM has attracted more attention recently with better robustness against noises and outliers [37]. Very recently, some methods [36, 50, 44] exploit the Neural Radiance Fields (NeRF) for dense visual SLAM in a rendering-based optimization framework showing appealing rendering quality in novel view. However, different from the traditional feature-based SLAM system, such learning-based methods with implicit representation are normally pure visual odometry systems without loop closure and pose graph optimization due to the limitation of the scene representation (e.g., a neural network or feature grid), which also makes it hard to be adapted to the collaborative SLAM. Take the feature grid representation as an example, it is hard to adjust or align the feature grid when the pose is optimized after loop closure or transformed to the unified global coordinate system for collaborative SLAM.

In this paper, we introduce a novel collaborative neural point-based SLAM system, named CP-SLAM, which enables cooperative localization and mapping for multiple agents and inherently supports loop closure for a single agent. However, it is nontrivial to design such a system. At first, we need a new neural representation for SLAM that is easy to be adjusted for loop closure and collaborative SLAM. Inspired by Point-NeRF [43], we built a novel neural point-based scene representation with keyframes.

---

*Corresponding author.

The scene geometry and appearance are encoded in 3D points with per-point neural features, and each point is associated with a certain keyframe. In this way, when the camera poses are optimized with loop closure and pose graph optimization, these neural points can be easily adjusted like traditional 3D points. Second, different from the monocular implicit SLAM system, a new learning strategy is required for collaborative implicit SLAM. For better consistency and cooperation, we present a two-stage learning strategy, i.e., distributed-to-centralized learning. In the first stage (before sub-map fusion), we set up an independent group of decoders for each RGB-D sequence and update them separately. In the second stage (after sub-map fusion), we fuse weights from all groups of decoders and continue with lightweight fine-tuning, after which all sequences can share and jointly optimize a common group of decoders. Furthermore, like the bundle adjustment in traditional SLAM, a novel optimization framework is needed to adjust both the camera poses and scene geometry for the neural implicit SLAM system. To this end, we introduce the pose graph optimization into the implicit SLAM system followed by a global map refinement.

Our contributions can be summarized as follows. At first, we present the first collaborative neural implicit SLAM system, i.e., CP-SLAM, which is composed of neural point based odometry, loop detection, sub-map fusion, and global refinement. Second, we propose a new neural point 3D scene representation with keyframes, which facilitates map fusion and adjustment. Furthermore, novel learning and optimization frameworks are proposed to ensure consistent and accurate 3D mapping for cooperative localization and mapping. We evaluate CP-SLAM on a variety of indoor RGB-D sequences and demonstrate state-of-the-art performance in both mapping and camera tracking.

## 2   Related Work

**Single-agent Visual SLAM**. With the development of unmanned intelligence, visual SLAM becomes an active field in the last decades. Klein et al. proposed a visual SLAM framework [15] that separates tracking and mapping into different threads. This framework is followed by most of current methods. Traditional single-agent visual SLAM uses filtering or nonlinear optimization to estimate the camera pose. Filtering based methods [27, 3, 23] are more real-time capable but less globally consistent. In contrast, optimization-based [28, 29, 31, 4, 8, 19] methods can make full use of past information. Meanwhile, multi-source information is widely incorporated into the SLAM system to improve specific modules, such as light-weight depth estimation [48, 42].

**Collaborative Visual SLAM**. Collaborative SLAM can be divided into two categories: centralized and distributed. CVI-SLAM [14], a centralized visual-inertial framework, can share all information in a central server and each agent outsources computationally expensive tasks. In centralized SLAM, the server manages all sub-maps, performs map fusion and global bundle adjustment, and feeds processed information back to each agent. This pipeline is reproduced in CCM-SLAM [33], where each agent is equipped with a simple visual odometry and sends localization and 3D point cloud to the central server. In terms of distributed systems, Lajoie et al. developed a fully distributed SLAM [18], which is based on peer-to-peer communication and can reject outliers for robustness. NetVLAD [1] is used in [18] to detect loop closures. In addition, [10] proposed the compact binary descriptor specifically for multi-agent system. Compared with traditional collaborative systems, our system can perform dense mapping with fewer neural points.

**Neural Implicit Representation**. Neural implicit field showed outstanding results in many computer vision tasks, such as novel view synthesis [24, 25, 38, 2], scene completion [5, 11, 30] and object modelling [6, 22, 40, 45, 46]. In recent study, some works [47, 41] attempted to reversely infer camera extrinsics from the built neural implicit field. Inspired by different representations of neural field including voxel grid [21] and point cloud [43], NICE-SLAM [50] and Vox-Fusion [44] chose voxel grid to perform tracking and mapping instead of a single neural network which is limited by expression ability and forgetting problem. Both of them are most related works to ours. Besides, vMAP [16] can efficiently model watertight object models in the absence of 3D priors. ESLAM [13] turns to tri-plane and Truncated Signed Distance Field (TSDF) to solve RGB-D SLAM. In addition to purely neural-based methods, some hybrid systems, such as NeRF-SLAM [32] and Orbeez-SLAM [7], construct the neural map but use traditional methods to estimate poses.

## 3   Method

The overview of our collaborative SLAM system is shown in Fig. 1. Given a set of RGB-D sequences, our system incrementally performs tracking and mapping based on neural point cloud representation for each agent (Section 3.1). We incorporate a learning-based loop detection module that extracts unique descriptors for 2D frames, and stitches sub-maps through high-quality loop constraints

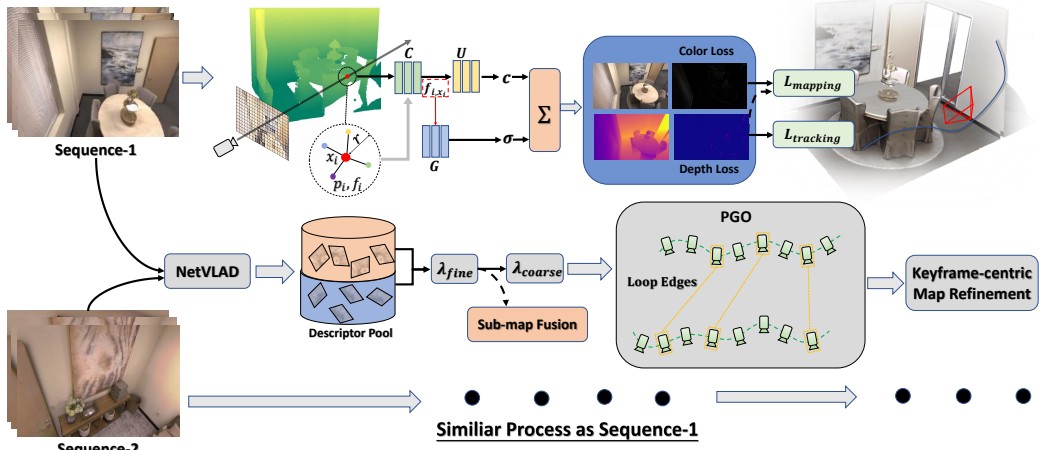

Figure 1: **System Overview.** Our system takes single or multi RGB-D streams as input and performs tracking and mapping as follows. From left to right, we conduct differentiable ray marching in a neural point field to predict depth and color. To obtain feature embedding of a sample point along a ray, we interpolate neighbor features within a sphere with radius $r$. MLPs decode these feature embeddings into meaningful density and radiance for volume rendering. By computing rendering difference loss, camera motion and neural field can be optimized. While tracking and mapping, a single agent continuously sends keyframe descriptors encoded by NetVLAD to the descriptor pool. The central server will fuse sub-maps and perform global pose graph optimization(PGO) based on matching pairs to deepen collaboration. Finally, our system ends the workflow with keyframe-centric map refinement.

(Section 3.2). We further design a two-stage (distributed-to-centralized) MLP training strategy to improve consistency and strengthen collaboration (Section 3.3). To reduce cumulative error in mapping and tracking, we use co-visibility among sub-maps to take global pose graph optimization as the back-end processing, followed by frame-based map refinement (Section 3.4). We will elaborate on the entire pipeline of our system in the following subsections.

## 3.1 Neural Point based Odometry

Our system starts from a front-end visual odometry module, in which we combine pose back-propagation update and point-based neural field to perform sequential tracking and mapping.

**Point-based Neural Field.** We divide an image into $4 \times 4$ patches and incrementally project each central pixel of every patch to its corresponding 3D location $p \in \mathbb{R}^3$. To acquire the feature embedding $f \in \mathbb{R}^{32}$ anchored on $p$, a 2D image is fed into an untrained single-layer convolutional neural network. We define our neural point cloud as:

$$P = \{p_j, f_j | j = 1, ..., N\}, \tag{1}$$

**Volume Rendering.** We emulate the differentiable volume rendering strategy in NeRF [26], where radiance and occupancy are integrated along rays to render color and depth maps. Given a 6DoF pose $\{\mathbf{R_c}, \mathbf{O_c}\}$ and intrinsics of a frame, we can cast rays from randomly selected pixels and sample $N_{total}$ points. Any sampling point is defined as

$$x_i = \mathbf{O_c} + t_i * \mathbf{d} \quad i \in 1...N_{total}, \tag{2}$$

where $t_i \in \mathbb{R}$ is the depth of the point and $\mathbf{d} \in \mathbb{R}^3$ is the unit ray direction. Guided by depth information, we can distribute our samples near the real surface. Specifically, we sample $N_{total} = N_{near} + N_{uni}$ points on each ray. For any pixel with valid depth $D$, $t_i$ is uniformly sampled within the intervals $[0.95D, 1.05D]$ and $[0.95D_{min}, 1.05D_{max}]$ respectively, yielding $N_{near}$ and $N_{uni}$ sample points correspondingly, where $D_{min}$ and $D_{max}$ are minimum and maximum depth values of the current depth map. For any pixel without valid depth values, $t_i$ is uniformly sampled within the interval $[D_l, 1.05D_{max}]$, generating $N_{total}$ points. For each point $x_i$, we firstly query $K$ neighbors $\{p_k | k = 1, ..., K\}$ within query radius $r$ and then use an MLP $C$ to convert the original neighbor feature $f_k$ into $f_{k,x_i}$ that incorporates relative distance information, i.e.,

$$f_{k,x_i} = C(f_k, x_i - p_k), \tag{3}$$

Then a radiance MLP $U$, decodes the RGB radiance $c$ at location $x_i$ using an interpolation feature $f_{x_i}$, which is obtained from weighted interpolation according to inverse distance weights and neighboring features:

$$f_{x_i} = \sum_k \frac{w_k}{\sum w_k} f_{k,x_i}, w_k = \frac{1}{\|p_k - x_i\|}, \tag{4}$$

$$c = U(f_{x_i}), \tag{5}$$

If no neighbors are found, the occupancy $\sigma$ at $x_i$ is set to zero. Otherwise, we regress $\sigma$ using an occupancy MLP $G$ at $x_i$. We follow a similar inverse distance-based weighting interpolation as radiance, instead of interpolating at feature level. We decode $\sigma_i$ for each neighbor and finally interpolate them:

$$\sigma_k = G(f_{k,x_i}), \tag{6}$$

$$\sigma = \sum_k \frac{w_k}{\sum w_k} \sigma_k, \tag{7}$$

Next we use $c_{x_i}, \sigma_{x_i}$ regressed from MLPs to estimate per-point weight $\alpha_{x_i}$. $\alpha_{x_i}$ is regarded as the opacity at $x_i$, or the probability of a ray terminates at this point and $z_{x_i}$ is the depth of point $x_i$. Depth map and color map can be rendered by calculating depth and radiance expectations along rays as in Eq. 8.

$$\hat{D} = \sum_{i=1}^{N_{total}} \alpha_{x_i} z_{x_i}, \quad \hat{I} = \sum_{i=1}^{N_{total}} \alpha_{x_i} c_{x_i}, \tag{8}$$

**Mapping and Tracking.** We use rendering difference loss as described in Eq. 9 that consists of geometric loss and photometric loss during the mapping process. For a new-coming frame, we sample $M_1$ pixels to optimize point-anchored features $f_i$ and parameters of MLP $C,U,G$. With the first frame, we need to perform a good initialization at a few higher cost of $M_3$ pixels and around 3000~5000 optimization steps to ensure smooth following processing. For subsequent mapping, we select pixels uniformly from the current frame and 5 co-visible keyframes. We find that joint mapping can effectively stabilize the tracking process.

$$\mathcal{L}_{mapping} = \frac{1}{M_1} \sum_{m=1}^{M_1} |D_m - \hat{D}_m| + \lambda_1 |I_m - \hat{I}_m|, \tag{9}$$

where $D_m$, $I_m$ represent ground truth depth and color map, $\hat{D}_m$, $\hat{I}_m$ are corresponding rendering results, and $\lambda_1$ is the loss balance weight. During the tracking process, we backpropagate farther to optimize camera extrinsics $\{q, t\}$ while keeping features and MLPs fixed, where $q$ is quaternion rotation and $t$ is the translation vector. We sample $M_2$ pixels across a new-coming frame and assume the zero motion model where the initial pose of a new frame is identical to that of the last frame. Considering the strong non-convexity of the color map, only the geometry part is included in tracking loss:

$$\mathcal{L}_{tracking} = \frac{1}{M_2} \sum_{m=1}^{M_2} |D_m - \hat{D}_m|, \tag{10}$$

### 3.2 Loop Detection and Sub-Map Alignment

To align sub-maps and reduce accumulated pose drift, we perform loop detection and calculate relative poses between different sub-maps. For each keyframe in the sub-map, we associate it with a descriptor generated by the pre-trained NetVLAD [1] model. We use cosine similarity between descriptors as the judgment criteria for loop detection. When the camera moves too much between frames, pose optimization tends to fall into local optimum, so we use two similarity thresholds $\lambda_{fine}, \lambda_{coarse}$ to find matching pairs that have enough overlap and ensure correct loop relative pose. Given two sub-maps, $\mathcal{M}_1$ and $\mathcal{M}_2$, we find out loop frames with largest overlap, $\{I^{l1}, I^{l2}\}$, with similarity greater than $\lambda_{fine}$, and small overlap loop frames, $\{(I_1^{s1}, I_2^{s2}), ..., (I_n^{s1}, I_n^{s2})\}$, with similarity greater than $\lambda_{coarse}$ but less than $\lambda_{fine}$. For pairs $\{I^{l1}, I^{l2}\}$, we take pose of $I^{l1}$ as initial pose and $I^{l2}$ as the reference frame. The tracking method described in 3.1 can be used to obtain the relative pose measurement between $\{\mathcal{M}_1, \mathcal{M}_2\}$. We can use this relative pose to perform a 3D

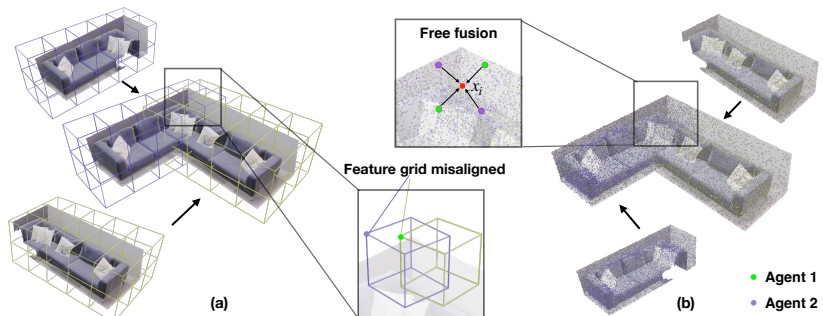

Figure 2: **Voxel Grid Limitation.** (a) If the initial coordinate system is not uniform, the neural field representations of existing neural SLAM such as voxel grid will suffer from misalignment during sub-map fusion. (b) The point cloud is not restricted by 3D geometry shape and can be fused freely. After sub-map fusion, We can query neighbors at $x_i$ from observations of Agent1 (green point cloud) and Agent2 (purple point cloud).

rigid transformation on sub-maps, so that all sub-maps are set in the same global coordinate system, achieving the purpose of sub-map alignment as shown in Fig. 2.

$$\mathcal{L}_{loop}(T_r) = \frac{1}{M_2} \sum_{\{I^{l1}, I^{l2}\}}^{M_2} |D_m(T_r) - \hat{D}_m|, \tag{11}$$

where $T_r$ is the relative pose between $\mathcal{M}_1$ and $\mathcal{M}_2$, and $D, \hat{D}$ is the rendered depth and ground truth depth. For loop frames with small overlap $\{I_i^{s1}, I_i^{s2}\}$, we only use them to perform pose graph optimization (refer to 3.4). Sub-map fusion will lead to neural point redundancy in specific areas, which imposes a burden on computing and memory. Due to the sparsity of the neural point cloud, we adopt a grid-based filtering strategy, that is, we perform non-maximum suppression based on the distance from a neural point to the center of a $\rho^3$ cube.

### 3.3 Distributed-to-Centralized Learning

To enhance consistency and cooperation, in collaborative SLAM, we adopt a two-stage MLP training strategy. At the first stage (Distributed Stage), each image sequence is considered as a discrete individual with a unique group of MLPs $\{C_j, U_j, G_j\}$ for sequential tracking and mapping. After loop detection and sub-map fusion, we expect to share common MLPs across all sequences (Centralized Stage). To this end, we introduce the Federated learning mechanism which

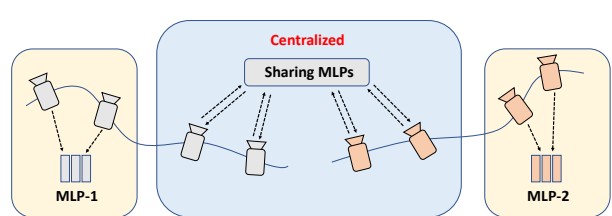

Figure 3: **Two-stage Learning Strategy.**

trains a single network in a cooperating shared way. At the same time as sub-map fusion, we average each group of MLPs and fine-tune the averaged MLPs on all keyframes to unify discrete domains. Subsequently, we iteratively transfer sharing MLPs to each agent for local training and average the local weights as the final optimization result of sharing MLPs, as shown in Fig. 3.

### 3.4 Pose Graph Optimization and Global Map Refinement

After front-end processing of all sequences, including tracking, mapping, loop detection, and sub-map fusion, we establish a global pose graph model, in which per-frame poses are nodes, sequential relative pose, and loop relative pose are edges. The pose graph model is illustrated in Fig. 4. We carry out global pose graph optimization across the entire pose graph, referring to traditional visual SLAM, to force the estimated trajectory closer to the ground truth. Global pose graph

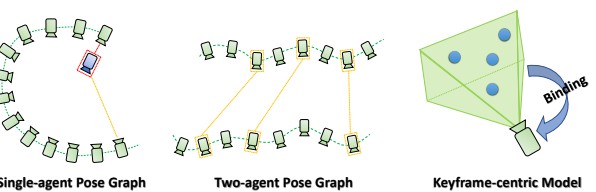

Figure 4: **Pose Graph and Keyframe-centric Model.**

optimization effectively mitigates cumulative error and improves tracking accuracy. We use the *Levenberg Marquarelt* algorithm to solve this nonlinear global pose graph optimization problem described by Eq. 12, where $v$ is the set of nodes, $E_s$ is the set of sequential edges, $E_l$ is the set of loop edges and $\Lambda_i$ represents the uncertainty of corresponding edges.

$$v^* = arg \min_v \frac{1}{2} \sum_{e_i \in E_s, E_l} e_i^T \Lambda_i^{-1} e_i, \tag{12}$$

Naturally, it is expected that the neural point cloud layout should be rearranged following global pose graph optimization. However, a world-centric point cloud map obviously cannot allow such adjustment. To tackle this limitation, we propose a keyframe-centric neural point field, where each 3D point is associated with a keyframe (Fig. 4). Benefitting from this design, we can refine our point cloud 3D locations according to optimized poses. Following global map refinement, we employ grid-based filtering to deal with neural point cloud redundancy occurring in local regions. Considering the slight mismatch between refined neural fields and decoders after global map refinement, we end up performing a low-cost fine-tuning of the global neural point fields with fewer optimization iterations.

## 4 Experiments

Our CP-SLAM system supports both single-agent and multi-agent modes. Thus, we evaluate our proposed collaborative SLAM system in two aspects, both single-agent experiments with loop closure and two-agent experiments, of varying sizes and complexity. In terms of a single agent, we generate datasets based on the Replica [35] scenes and then compare our method against recent neural and traditional RGB-D SLAM methods. For the two-agent side, since no collaborative neural SLAM work has emerged so far, we compare our method with traditional methods. We also conduct ablation studies to show the importance of modules in the proposed system.

**Implementation Details.** CP-SLAM system runs an RGB-D sequence on an NVIDIA RTX3090 GPU. In the two-agent experiment, we need an additional RTX3090 as the central server. To encode higher frequency detail information, during inference, we impose position encoding on the origin neighbor feature $f_i$ and relative distance $\|p_i - x\|$ with an order of 1 and 7. We utilize the FRNN library to query $K = 8$ nearest neighbors on GPU. In all our experiments, we set $N_{near} = 16, N_{uni} = 4, \lambda_1 = 0.2, D_l = 0.001m, r = 0.15m, \rho = 0.14m, M_1 = 3000, M_3 = 3136, M_2 = 1500$. We extract a keyframe every 50 frames and perform map optimization and point cloud supplementation every 10 frames. For single-agent experiments, we optimize the neural field for 200 iterations. For two-agent experiments, considering that features anchored on neural points do not need to be trained from scratch after sub-map fusion, we reduce the number of iteration steps to 150. Further implementation details can be found in our supplementary material.

**Baselines.** In the single-agent experiment, because we use the rendered loop-closure data, we primarily choose the state-of-the-art neural SLAM systems such as NICE-SLAM [50], Vox-Fusion [44] and ORB-SLAM3 [4] for comparison on the loop-closure dataset. For the two-agent experiment, we compare our method with traditional approaches, such as CCM-SLAM [33], Swarm-SLAM [17] and ORB-SLAM3 [4].

**Datasets.** For reconstruction assessment, we utilize the synthetic dataset Replica [35], equipped with a high-quality RGB-D rendering SDK. We generate 8 collections of RGB-D sequences, 4 of which represent single-agent trajectories, each containing 1500 RGB-D frames. The remaining 4 collections are designed for collaborative SLAM experiments. Each collection is divided into 2 portions, each holding 2500 frames, with the exception of Office-0-C which includes 1950 frames per part.

**Metrics.** As a dense neural implicit SLAM system, we quantitatively and qualitatively measure its mapping and tracking capabilities. For mapping, we evaluate L1 loss between 196 uniformly-sampled depth maps, which are rendered from our neural point field, and ground truth ones. Furthermore, in terms of 3D triangle mesh, we compute mesh reconstruction accuracy. For tracking, we use ATE RMSE, Mean and Median to comprehensively measure trajectory accuracy so as to prevent the negative impact caused by a few extreme outliers.

### 4.1 Two-agent Collaboration

We provide the quantitative results of two-agent experiments on four scenes including Replica [35] Apartment-0, Apartment-1, Apartment-2 (multi-room), and Office-0-C (single-room). We take RGB-based CCM-SLAM [33], RGBD-based Swarm-SLAM [17] and traditional ORB-SLAM3 [4] for comparison. Table. 1 reports the localization accuracy of different methods. Despite being affected

| Method | Part | Apartment-1 | Apartment-2 | Apartment-0 | Office-0-C (Single Room) |
|---|---|---|---|---|---|
| | | RMSE[cm]↓ / Mean [cm]↓ / Median [cm]↓ | | | |
| CCM-SLAM [33] | | 2.12/1.94/1.74 | **0.51/0.45/0.40** | -/-/- | 9.84/8.23/6.41 |
| ORB-SLAM3 [4] | | 4.93/4.65/5.01 | 1.35/1.05/0.65 | 0.67/0.58/0.47 | 0.66/0.62/0.62 |
| Swarm-SLAM [17] | **Part 1** | 4.62/4.17/3.90 | 2.69/2.48/2.34 | 1.61/1.33/1.09 | 1.07/0.96/0.98 |
| Ours (w/o) | | 1.15/0.99/0.88 | 1.45/1.34/1.36 | 0.70/0.48/**0.27** | 0.71/0.62/0.67 |
| Ours (w/) | | **1.11/0.95/0.81** | 1.41/1.30/1.36 | **0.62/0.47**/0.30 | **0.50/0.46/0.55** |
| CCM-SLAM | | 9.31/6.36/5.57 | **0.48/0.43/0.38** | -/-/- | 0.76/**0.36/0.16** |
| ORB-SLAM3 | | 4.93/4.04/3.80 | 1.36/1.24/1.11 | 1.46/1.11/0.79 | **0.54**/0.49/0.47 |
| Swarm-SLAM | **Part 2** | 6.50/5.27/4.39 | 8.53/7.59/7.10 | 1.98/1.48/0.94 | 1.76/1.55/1.83 |
| Ours (w/o) | | 2.12/2.05/2.23 | 2.54/2.45/2.60 | 1.61/1.55/1.70 | 1.02/1.03/0.99 |
| Ours (w/) | | **1.72/1.61/1.46** | 2.41/2.33/2.44 | **1.28/1.17/1.37** | 0.79/0.74/0.70 |
| CCM-SLAM | | 5.71/4.15/3.66 | **0.49/0.44/0.39** | -/-/- | 5.30/4.29/3.29 |
| ORB-SLAM3 | | 4.93/4.35/4.41 | 1.36/1.15/0.88 | 1.07/0.85/**0.63** | **0.60/0.56/0.55** |
| Swarm-SLAM | **Average** | 5.56/4.72/4.15 | 5.61/5.04/4.72 | 1.80/1.41/1.02 | 1.42/1.26/1.41 |
| Ours (w/o) | | 1.64/1.52/1.56 | 2.00/1.90/1.98 | 1.16/1.02/0.99 | 0.86/0.81/0.83 |
| Ours (w/) | | **1.42/1.28/1.14** | 1.91/1.82/1.90 | **0.95/0.82**/0.84 | 0.65/0.60/0.63 |

Table 1: **Two-agent Tracking Performance.** ATE RMSE(↓), Mean(↓) and Median(↓) are used as evaluation metrics. We quantitatively evaluated respective trajectories (part 1 and part 2) and average results of the two agents. Comparison between ours(w/o) and ours(w/) reveals the importance of global pose graph optimization for collaborative tracking. "-" indicates invalid results due to the failure of CCM-SLAM.

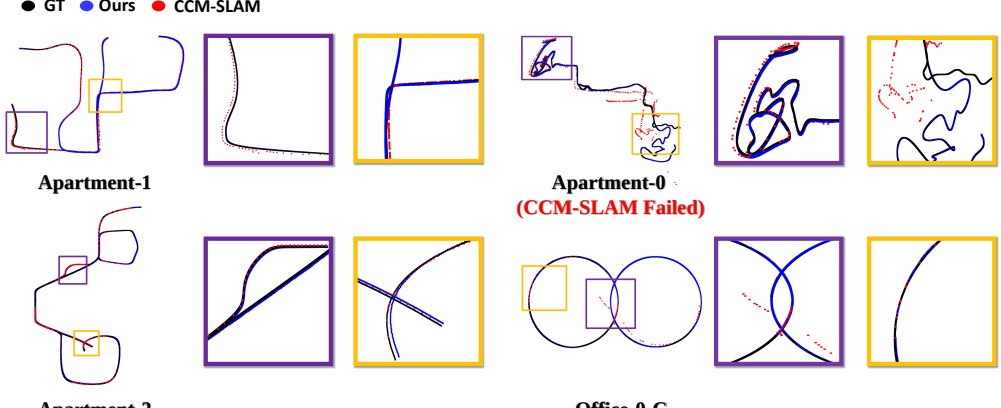

Figure 5: **Two-agent Trajectories on 4 Scenes.** CCM-SLAM relies too much on 2D geometric information so that it has a large drift. In contrast, our neural implicit system has a robust performance.

by complex environments in multi-room sequences, the proposed system generally maintains better performance than other methods. It is worth noting that ORB-SLAM3 is not a collaborative SLAM system. It lacks the capability to process multiple sequences at the same time, thus unable to overcome the efficiency bottleneck in scene exploration. Specifically, we concatenate multiple image sequences and feed them into ORB-SLAM3, leveraging its "atlas" strategy for multi-sequence processing. Also, it can be seen that CCM-SLAM failed in some scenes because traditional RGB-based methods are prone to feature mismatching especially in textureless environments. Fig. 5 depicts the trajectories of each agent. Once two sub-graphs are fused, only a low-cost fine-tuning is required to adjust two neural fields and corresponding MLPs into a shared domain. Afterward, these two agents can reuse each other's previous observations and continue accurate tracking. Shared MLPs, neural fields, and following global pose graph optimization make CP-SLAM system a tightly collaborative system.

## 4.2 Single Agent with Loop

In Table. 2, we illustrate the localization performance of our system operating in single-agent mode on 4 loop closure datasets. Our method exhibits a notable superiority over recent methods, including NeRF-based and traditional ones, primarily attributed to the integration of concurrent front-end and back-end processing, as well as the incorporation of neural point representation. Qualitatively, we present trajectories of Room-0-loop and Office-3-loop from NeRF-based methods in Fig. 6. The

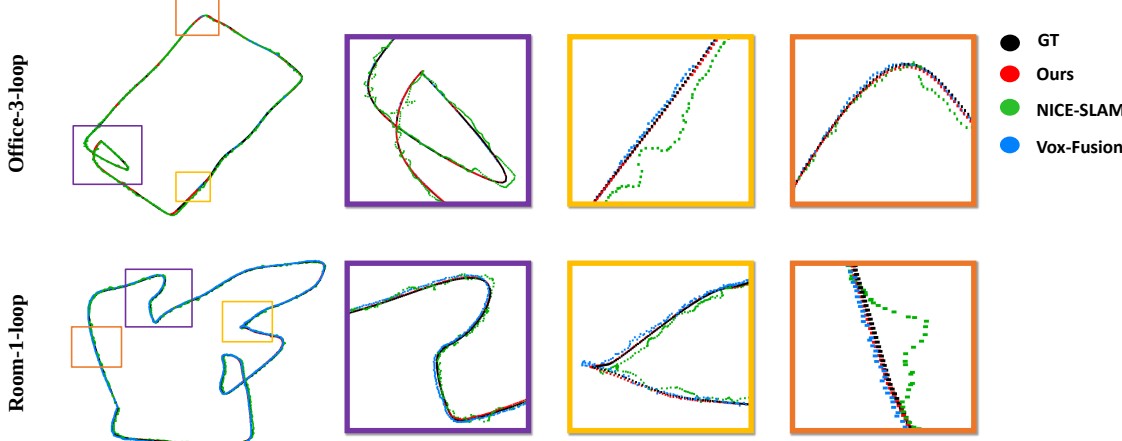

Figure 6: **Single-agent Trajectories on 4 Scenes.** In comparison with frequent jitters in the trajectories of the other two methods, our trajectory is much smoother.

| Method | Metric | Room-0-loop | Room-1-loop | Office-0-loop | Office-3-loop | Average |
|---|---|---|---|---|---|---|
| NICE-SLAM [50] | RMSE [$cm$] ↓ | 1.27 | 1.74 | 2.27 | 3.19 | 2.12 |
| | Mean [$cm$] ↓ | 1.15 | 1.61 | 1.91 | 2.77 | 1.86 |
| | Median [$cm$] ↓ | 1.09 | 1.66 | 1.82 | 2.28 | 1.71 |
| Vox-Fusion [44] | RMSE [$cm$] ↓ | 0.82 | 1.35 | 0.99 | 0.82 | 0.99 |
| | Mean [$cm$] ↓ | 0.77 | 1.30 | 0.94 | 0.74 | 0.94 |
| | Median [$cm$] ↓ | 0.78 | 1.25 | 0.95 | 0.73 | 0.93 |
| ORB-SLAM3 [4] | RMSE [$cm$] ↓ | 0.54 | **0.21** | 0.58 | 0.89 | 0.56 |
| | Mean [$cm$] ↓ | 0.52 | **0.19** | 0.51 | 0.80 | 0.51 |
| | Median [$cm$] ↓ | 0.53 | **0.19** | 0.52 | 0.84 | 0.52 |
| Ours(w/o) | RMSE [$cm$] ↓ | 0.61 | 0.51 | 0.67 | 0.38 | 0.54 |
| | Mean [$cm$] ↓ | 0.56 | 0.48 | 0.63 | 0.32 | 0.50 |
| | Median [$cm$] ↓ | 0.54 | 0.52 | 0.67 | **0.27** | 0.50 |
| Ours (w/) | RMSE [$cm$] ↓ | **0.48** | 0.44 | **0.56** | **0.37** | **0.46** |
| | Mean [$cm$] ↓ | **0.44** | 0.40 | **0.53** | **0.31** | **0.42** |
| | Median [$cm$] ↓ | **0.43** | 0.46 | **0.56** | **0.27** | **0.43** |

Table 2: **Single-agent Tracking Performance.** Our system consistently yields better results compared with existing single-agent neural and traditional SLAM. In the fourth and fifth rows, we compare the accuracy of our system without and with the pose graph optimization module. Results are obtained in an origin-aligned manner with the EVO [9] toolbox.

experimental results demonstrate that the concentration level of density energy around sampling points is critical for neural SLAM. NICE-SLAM [50] uses dense voxel grids, which contain a large number of empty spaces, and sampling points in these empty spaces have almost no contribution to the gradient propagation. Vox-Fusion [44] has incorporated an important modification, implementing a sparse grid that is tailored to the specific scene instead of a dense grid. However, grid nodes can only be roughly placed near objects, which is unfavorable for tracking. In our point-based method, we ensure that neural field fits the real scene well. Feature embeddings can accurately include scene information. Concentratively distributed sample points and neural point based representation provide more exact gradients for pose backpropagation, which enables us to surpass NICE-SLAM and Vox-Fusion at lower resolution and memory usage. Moreover, we extend experiments on the TUM-RGBD real-world dataset, comparing with Co-SLAM [39] and ESLAM [12]. The results in Table. 3 illustrate that our method has also achieved state-of-the-art performance in the real-world setting, and the loop detection and pose graph optimization are equally effective for the real-world scene.

## 4.3 Map Reconstruction

The results in Table. 4 present a quantitative analysis of the geometric reconstruction produced by our proposed system in comparison to NICE-SLAM [50] and Vox-Fusion [44]. In our approach, we render depth maps and color maps every 10 frames throughout the entire predicted trajectories and utilize TSDF-Fusion (built-in function in Open3D [49] library) to construct mesh map. In

| Method | fr1-desk (w/o loop) | fr2-xyz (w/o loop) | fr3-office (w/ loop) | Average |
|---|---|---|---|---|
| Co-SLAM [39] | 7.10/6.83/6.79 | 4.05/3.76/3.47 | 5.58/5.06/4.57 | 5.58/5.22/4.95 |
| ESLAM [12] | **6.81/6.56/6.88** | Fail/Fail/Fail | 4.23/3.91/3.73 | - |
| Ours | 7.84/7.34/7.12 | **3.93/3.50/3.29** | **3.84/3.47/3.39** | **5.20/4.77/4.60** |

Table 3: **Real-world Tracking Performance.** In this real-world experiment, we can find that CP-SLAM still performs the best. Besides, ESLAM fails in the fr2-xyz because of OOM (out of memory). '-' indicates that metrics cannot be evaluated due to ESLAM failures.

| Method | Metric | Room-0-loop | Room-1-loop | Office-0-loop | Office-3-loop | Average |
|---|---|---|---|---|---|---|
| NICE-SLAM [50] | Depth L1 $[cm]\downarrow$ | 1.54 | 1.00 | 0.93 | 2.06 | 1.38 |
| | Acc. $[cm]\downarrow$ | 3.30 | 3.19 | 2.88 | 3.95 | 3.33 |
| Vox-Fusion [44] | Depth L1 $[cm]\downarrow$ | 0.77 | 1.30 | 0.94 | **0.74** | 0.94 |
| | Acc. $[cm]\downarrow$ | 2.25 | 1.67 | 1.68 | 2.31 | 1.98 |
| Ours | Depth L1 $[cm]\downarrow$ | **0.32** | **0.23** | **0.22** | 0.76 | **0.38** |
| | Acc. $[cm]\downarrow$ | **1.53** | **1.20** | **1.21** | **1.7** | **1.41** |

Table 4: **Reconstruction Results.** Our system has a more powerful geometric reconstruction capability than existing methods.

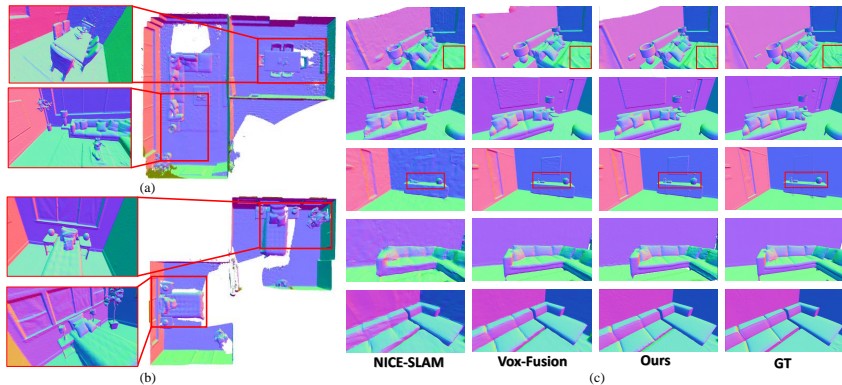

Figure 7: **Reconstruction.** Collaborative reconstruction results of Apartment-1 (a) and Apartment-0 (b) show good consistency. (c) It can be seen that our system also achieves more detailed geometry reconstruction in four single-agent datasets, e.g., note folds of the quilt and the kettle in red boxes. Holes in the mesh reconstruction indicates unseen area in our datasets.

NICE-SLAM, several metrics are employed for mapping evaluation, namely depth L1 loss, mesh accuracy, completion, and completion ratio. In our loop datasets, scenes are not completely scanned, which leads to holes in mesh reconstruction. Therefore, our comparative experiments mainly focus on depth L1 loss and mesh accuracy. Fig. 7 qualitatively compares the single-agent reconstruction results of three methods and shows our collaborative mapping results. Evidently, our method achieves more detailed geometry across all datasets.

## 4.4 Ablation Study

In this section, we examine some modules and designs in our system to prove their importance and the rationality of our pipeline.

**Pose Graph Optimization.** In this section, we conducted ablation experiments on the PGO module. Table. 1 and Table. 2 report results of ours(w/o PGO) and ours(w/ PGO) in single-agent and two-agent cases respectively. With the help of PGO module, the average positioning accuracy in the single-agent experiment decreases by 10%, while that in the two-agent experiment decreases by 13%.

**Map Refinement.** As shown in Fig. 8(a), we qualitatively illustrate the neural point field layout before and after map refinement in MeshLab. We can observe that a refined neural point cloud fits the ground truth mesh better.

**Sampling Concentration.** The density concentration near sampling points is a keypoint that determines the performance of neural SLAM. We design a set of experiments with different sampling interval lengths on Replica Room-0-Loop sequence. As shown in Fig. 8(b), tracking accuracy and depth L1 loss consistently drop to 1.0cm and 0.6cm as sampling points gradually diverge. This experiment fully verifies the theory in Section. 4.2.

**Neural Point Density.** In CP-SLAM, we employed a fixed-size cubic cell within the filtering strategy

|  | Metric | $\rho$=10cm | $\rho$=14cm | $\rho$=18cm | $\rho$=22cm |
|---|---|---|---|---|---|
|  | RMSE [$cm$] ↓ | 0.83 | **0.65** | 0.86 | 1.12 |
| Ours (w/) | Mean [$cm$] ↓ | 0.77 | **0.58** | 0.76 | 1.01 |
|  | Median [$cm$] ↓ | 0.83 | **0.51** | 0.71 | 0.94 |

Table 5: **Neural Point Density Analysis.** Results indicate that the point cloud density should be at an appropriate level, neither too high nor too low. Empirically, we have found that our system achieves the best performance when the cubic cell is set to $\rho = 14cm$.

| Method | Tracking/Frame | Mapping/Frame | MLP Size | Feature Size |
|---|---|---|---|---|
| NICE-SLAM [50] | 1.77s | 11.26s | **0.58$\times10^{\wedge}5$** | 238.88MB |
| Vox-Fusion [44] | 0.36s | **0.83s** | 2.73$\times10^{\wedge}5$ | **0.15MB** |
| Ours | **0.30s** | 10.10s | 1.36$\times10^{\wedge}5$ | 0.62MB |

Table 6: **Runtime and Memory Analysis.**

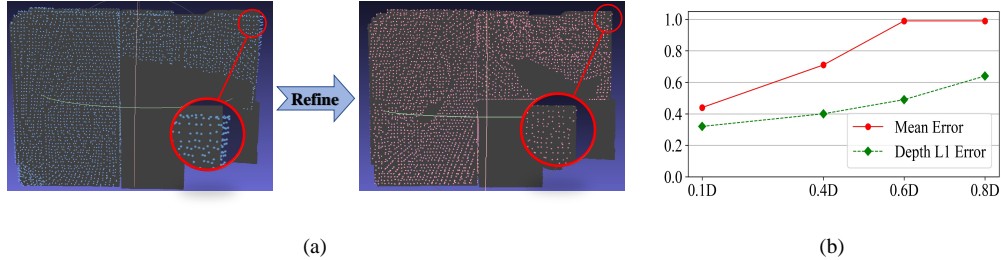

(a)                    (b)

Figure 8: **Map Refinement and Sampling Concentration Ablation.** (a) From a bottom view of Apartment-1, we can observe that there is an offset in the unrefined neural point cloud(blue), while the refined neural point cloud(pink) fits well. (b) As the sampling points diverge, the error rises.

to adjust the neural points, i.e., only the neural point closest to the center of a cubic cell is retained. To further explore the impact of point cloud density on tracking accuracy, we compared performance in the original Replica Room0 scene across various sizes of the cubic grid. The ablation study results in Table. 5 demonstrate that when the number of neural points is small, they are not enough to encode detailed scene geometry and color due to inadequate scene representation. Conversely, an excessive number of neural points obviously extends the learning time to converge. We empirically found that the setting $\rho = 14cm$ worked consistently well on all the test scenes with different complexities in our experiment.

**Memory and Runtime Analysis.** We evaluated the runtime and memory consumption of our system on Replica Office-0-loop scene compared to NICE-SLAM and Vox-Fusion. We report single-frame tracking and mapping time, the size of MLPs and memory footprint of the whole neural field in Table. 6. The huge feature size in NICE-SLAM is due to its dense hierarchical feature grid.

## 5   Conclusion

We have proposed CP-SLAM, the first dense collaborative neural SLAM framework based on a novel neural point based representation, which maintains complete front-end and back-end modules like traditional SLAM systems. The comprehensive pipeline enables our system to outperform the state-of-the-art methods in both localization and reconstruction. One limitation of our method is its requirement for considerable GPU resources to operate multiple image sequences. Also, our system has slightly weaker hole-filling ability in unobserved regions than feature grid-based methods, which arises from the fact that neural points are distributed around the surfaces of observed objects, encoding surrounding scene information within a fixed-radius sphere. Moreover, the relative pose computation in the loop closure relies on the existing rendering-based optimization, which may be inaccurate for large viewpoint changes thus leading to drifting of map fusion. Hence, it is interesting to design a lightweight system with a coarse-to-fine pose estimation for future work.

**Acknowledgment:** This work was partially supported by the NSFC (No. 62102356).

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
