# OpenReview forum: "CP-SLAM: Collaborative Neural Point-based SLAM System"
_NeurIPS.cc/2023/Conference — NeurIPS 2023 poster_

### Official Review · Reviewer_jr2H · 2023-07-05

**Soundness:** 3 good
**Presentation:** 3 good
**Contribution:** 3 good
**Rating:** 6
**Confidence:** 4

**Summary:**

The paper presents a collaborative implicit neural simultaneous localization and mapping (SLAM) system for RGB-D image sequences, which is able to merge sub-maps into a global map and also handle loop-closure for neural mapper for the first time. To facilitate the capability of doing loop-closure and sub-map merging, the authors introduce a neural point-based 3D scene representation, where each point maintains a learnable neural feature for scene encoding and is associated with a keyframe. The optimization follows the traditional SLAM’s graph optimization fashion. The experimental results on diverse datasets demonstrate the superior performance of the proposed method in both camera tracking and mapping.

**Strengths:**

1. Novel Neural Point-based 3D Scene Representation: A neural point cloud representation for scene encoding in the proposed framework is a significant contribution. This approach allows each point to carry encoded information, improving the system's ability to represent and cross-view association for large-scale tracking and pose refinement.
2.Comprehensive Collaborative Framework: The authors provide a collaborative SLAM system with front-end and back-end modules, addressing essential aspects such as odometry, loop detection, sub-map fusion, and global refinement. This comprehensive approach ensures that the proposed system can handle various challenges typically encountered in SLAM tasks.
3.Distributed-to-Centralized Learning Strategy: The incorporation of a distributed-to-centralized learning strategy is an effective solution for improving consistency and cooperation in collaborative implicit SLAM.
4.Improved Performance: The experimental results on multiple datasets demonstrate the superiority of the proposed method in both camera tracking and mapping.

**Weaknesses:**

1.In the Introduction: This paper emphasizes the significance of targeting collaborative-SLAM, considering the substantial value already provided by loop-closure capability in effectively reducing pose drift. However, the evaluation of the paper becomes more challenging due to the necessity for additional comparative studies that can convincingly demonstrate the advantages offered by both approaches.

2.In Section 2: For collaborative mapping, this paper should cite more recent works including:
[1*] Campos, Carlos, et al. "Orb-slam3: An accurate open-source library for visual, visual–inertial, and multimap slam." IEEE Transactions on Robotics 37.6 (2021): 1874-1890.
[2*] Tian, Yulun, et al. "Kimera-multi: Robust, distributed, dense metric-semantic slam for multi-robot systems." IEEE Transactions on Robotics 38.4 (2022).

It is also expected to provide comparative studies to the above researches.

3.In Section 3:
(1). Line 101: This paper introduced 4x4 patches to compute the point features. However, it misses details on there is a limitation on the number of patches. What if we increase the size of the patches? Will it impact the later performance?
(2) This paper introduced so many symbols but without giving any explanation of the symbols, which makes it really hard to follow the paper. Please revise the whole paper and make them clear to the readers.
(3) Equation 10: The explanation “Considering the strong non-convexity of the color map, only the geometry part is included in tracking loss” for not using color in the loss is not convincing. The authors are expected to provide ablation studies to prove the claim.

4.In Section 4: More experiments are expected:
(1). More comparative studies on Collaborative-SLAM are expected.
(2). For single-agent cases: The local covisibility map can be formed as well; it would be interesting to only look at cases without a loop but having a local covisibility map.
(3). Ablation study on “Distributed-to-Centralized Learning” is needed to prove why this is needed.

**Questions:**

1.In the context of this paper, which relies on point features for establishing frame-to-frame correspondence, an important concern arises regarding the maintenance of continuous tracking. Unlike the traditional SLAM pipeline, where feature locations typically reside around corners, this paper selects points without necessarily adhering to this characteristic. Thus, it becomes crucial to address how this approach ensures the continuity of tracking throughout the SLAM process.
2.Collaborative mapping can be done by recent work on No Pose Nerf:
[]Bian, Wenjing, et al. "Nope-nerf: Optimising neural radiance field with no pose prior." Proceedings of the IEEE/CVF Conference on Computer Vision and Pattern Recognition. 2023.
This paper should provide a comparison to show the advantage of the proposed method.
3. This paper has too many contributions, which does not form a scientific paper but more an engineering paper. Not sure if this paper fall into the NeuralIPS group.

**Limitations:**

1. The writing should be improved, especially the equations should be better explained.
2. More experiments and ablation studies should be provided to demonstrate the advantage of the proposed algorithms.
3. The literature review needs to refer to more recent research.

---

> ### Author Rebuttal · Authors · 2023-08-09
>
> Thank you for your careful review and valuable suggestions. Here are our responses to your comments.
>
> **1. To demonstrate advantages from both collaboration and loop-closure capability, evaluation in this paper becomes more challenging. (W1)**
>
> In CP-SLAM, loop detection and pose graph optimization (PGO) work together to mitigate pose drift, and collaboration mainly helps to improve exploration efficiency, but has little effect on pose drift. In Tab.2 and Tab.1, we perform comparative studies in  the **‘w/ and w/o PGO‘** setting and observed consistent improvement across all datasets. Overall these two experiments consistently demonstrate the effectiveness of our proposed pipeline.
>
> **2. This paper should cite and compare with more recent works, such as ORB-SLAM3 and Kimera-multi. (W2)**
>
> As far as we know, ORB-SLAM3 is not a collaborative SLAM system. It lacks the capability to process multiple sequences at the same time, thus unable to overcome the efficiency bottleneck in scene exploration. As suggested, we tested ORB-SLAM3 for a comprehensive comparison. We have sequentially connected multiple parts from a single multi-agent dataset. We reported multi-agent and single-agent results in **Tab.G1** and **Tab.G2** in global response. In addition, as a visual-inertial method, Kimera-multi takes IMU data as input, thus we did not consider it as a comparable method.
>
> **3. In Section 3: (1). Line 101: How the performance is affected by the number of patches? (2) Some symbols are not clear. (3) Equation 10: Authors should perform an ablation study to confirm their point on non-convexity from color loss. (W3)**
>
> **(1)** In principle, the number of patches will not affect CP-SLAM in terms of memory usage and accuracy. Regarding memory usage, more patches bring more 3D neural points. However, we apply a filtering strategy to maintain point cloud sparsity, as indicated in **Supp.D.** From the perspective of accuracy, we empirically found that an increased quantity of neural points slightly decreases the tracking and mapping performance due to the increased complexity and redundancy of the neural point field.
>
> **(2)** Thank you for your comment, we will check and clarify all symbols to make our paper more readable.
>
> **(3)** We conducted an additional ablation study to confirm our viewpoints, as shown in **Fig.3** in the attached **'pdf'** in global response.
>
> **4. (1). More comparisons on collaborative SLAM are expected. (2). For single-agent cases: it would be interesting to take a local covisibility map into account. (3). Ablation study on “Distributed-to-Centralized Learning” is needed to prove its necessity. (W4)**
>
> **(1)** We added ORB-SLAM3 and Swarm-SLAM as new baselines for fully comparative studies. The results are shown in **Tab.G1** in global response, we can observe that our method still performs the best.
>
> **(2)** In CP-SLAM, we have used covisible correlations between keyframes. Specifically, we utilize multiple frames, including the current frame, the nearest keyframe, and three other covisible keyframes, for joint mapping, which ensures the consistency of the neural field. However, CP-SLAM which relies on neural features is completely different from the traditional visual SLAM that relies on the point cloud. In the volume rendering-based SLAM, the local covisibility map cannot replace the loop detection module. We present results for single-agent cases with joint mapping but without PGO in Tab.2.
>
> **(3)** In **Supp.C**, We discussed the necessity of `Distributed-to-Centralized' learning strategy. We have attempted to perform centralized learning from scratch, but its results were remarkably poor.
>
> **5. It is crucial to address how CP-SLAM ensures the continuity of tracking throughout the SLAM process. (Q1)**
>
> CP-SLAM is a collaborative NERF-based SLAM system, similar to NICE-SLAM, relying on volume rendering to optimize poses and the neural field. Neural points serve solely as spatial carriers of neural features in CP-SLAM, which are different from traditional keypoints and descriptors, similar to feature grid nodes in NICE-SLAM. In CP-SLAM, we do not perform inter-frame feature matching, so there is no discontinuity in tracking and mapping due to feature locations. We provide a detailed video in our supplementary material, which dynamically shows NERF-based tracking and mapping.
>
> **6. This paper should consider taking Nope-Nerf as a comparison. (Q2)**
>
> Nope-nerf was not public before our paper was submitted. Furthermore, it is worth noting that Nope-nerf is more biased towards Structure-from Motion (SFM) rather than SLAM. It has a different background setting from our CP-SLAM, thus we didn't consider it as a comparable method.
>
> **7. Contributions of this paper are more biased towards engineering. (Q3)**
>
> We aim to build a complete collaborative NERF-based SLAM system, focusing more on system-level development. Such work is missing in the field of NERF-based SLAM. CP-SLAM is the first collaborative neural implicit SLAM system, consisting of neural point-based odometry, loop detection, sub-map fusion, and global pose graph optimization. In collaborative NERF-based SLAM, in addition to the lack of appropriate neural field representation, it is not feasible to simply stack and cascade traditional modules in a plug-and-play manner. Consequently, we have introduced technical enhancements to both the neural field representation and the pipeline of CP-SLAM, such as innovatively proposing a keyframe-centric neural point field and a novel learning framework to maintain global consistency. Overall, we have made significant scientific contributions to the up-to-date NERF-based SLAM field as Reviewer Zpdr pointed out. Similar to Droid-SLAM [Teed et al. NeurIPS 2021], we also committed to a system-level development and introducing new learning-based neural implicit techniques into SLAM.

---

> > ### Comment · Reviewer_jr2H · 2023-08-13
> > **2nd round comments**
> >
> > Thank you to the authors for their responses. The authors largely addressed my last comments.
> >
> > But there are still several more questions:
> > 1. After reading the responses, the reviewer still believes that the authors should separate it into two papers: 'Point-SLAM' and 'Collaborative Mapping.' Alternatively, the authors have to highlight the motivation in the Abstract and Introduction. Otherwise, it is really hard to understand why the authors want to include Collaborative Mapping in this paper.
> >
> > 2. The authors answered, 'From the perspective of accuracy, we empirically found that an increased quantity of neural points slightly decreases the tracking and mapping performance due to the increased complexity and redundancy of the neural point field.' This contradicts the conventional feature-based tracking SLAM approach. This will make the proposed method difficult to generalize and use.
> >
> > I hope the authors can address my concerns and I would like to increase my rating.

---

> > > ### Author Response · Authors · 2023-08-14
> > > **Response to Reviewer jr2H - 2nd round comments**
> > >
> > > Dear Reviewer jr2H:
> > >
> > > Thank you for reviewing our last responses. The following are our responses to the further comments.
> > >
> > > 1. **Motivation of Our paper and Clarification on Collaborative Mapping and Point-based NeRF SLAM**
> > >
> > >    We apologize for the misunderstanding regarding the initial question. Here, we want to explain our motivation and clarify our collaborative mapping and point-based NeRF SLAM.
> > >
> > >    In this paper, we aim to build a collaborative NeRF-based SLAM system instead of a new 'Point-SLAM' system. We find that the neural field representation used in existing works, like the feature grid in NICE-SLAM, cannot support the fusion of multiple neural fields that is essential for multi-agent collaboration,  since it is difficult to fuse the 'collision areas' among different feature grids as shown in Fig. 2 in our main paper. Thus, motivated by the Point-Nerf [Xu et al. 2022], we designed a novel keyframe-centric neural point-based representation for our collaborative NeRF-based SLAM in which each point associated with a certain keyframe maintains a learnable neural feature for scene encoding. This new representation enables the seamless fusion of different neural sub-maps for multi-agent mode, and it also natually facilitates the loop-closure for single-agent mode. Consequently, we conducted experiments under both multi-agent and single-agent scenarios. The former demonstrates the effectivenss of our full collabative SLAM system, while the latter shows the advantage from our new representation in monocular NeRF SLAM. We will make this clearer in our revised verison.
> > >
> > > 2. **The impact of the number of neural points**
> > >
> > >    As mentioned in our response to **'Q1'**, in CP-SLAM, neural points are utilized as spatial carriers for neural features, which are subsequently decoded by MLPs to generate color and depth through volume rendering. Consequently, when the number of neural points is small, they are not enough to encode detailed scene geometry and color due to inadequate scene representation.
> > >
> > >    Conversely, an excessive number of neural points significantly extends the learning time to converge (i.e., increases the number of optimization steps) for both neural point features and MLPs. This means that, within limited optimization steps, it will decrease the tracking and mapping performance due to the increased complexity and redundancy of the neural point field.
> > >
> > >    Bearing these insights in mind, we opted for 4\*4 patches to produce initial dense neural points, which yields a sufficient number of neural points to represent the complex scene. Furthermore, together with our proposed point filtering strategy (as discussed in **Sec.D** of our supplementary material), it achieves a good trade-off between efficiency and accuracy. It's worth noting that we consistently employed 4*4 patches for all experiments presented in the paper as well as for the new TUM RGB-D experiment, which demonstrates the generalization of our system.

---

> > > > ### Comment · Reviewer_jr2H · 2023-08-20
> > > > **Raise my rating**
> > > >
> > > > Thank you for the author's response to my comments.
> > > >
> > > > I have one more question regarding 'The impact of the number of neural points': Should the adjustment of the point count be adaptable to the scene's complexity? If so, you should contemplate incorporating an ablation study.
> > > >
> > > > This paper holds significant interest, and I am inclined to raise my rating from 5 to 6.

---

> > > > > ### Author Response · Authors · 2023-08-21
> > > > > **Response to Reviewer jr2H - Adjustment of the Neural Point Count**
> > > > >
> > > > > Dear Reviewer jr2H：
> > > > >
> > > > > Thank you for raising your rating and valuable comments. Here are our responses to your latest comments.
> > > > >
> > > > > In CP-SLAM, we employed a fixed-size cubic cell ($\rho=14cm$) within the filtering strategy to adjust the neural points like Point-NeRF, i.e., only the neural point closest to the center of a cubic cell is retained. We found that this setting worked consistently well on all the test scenes with different complexities in our experiment. However, it is indeed interesting to further explore the impact of the size of the cube on the final performance for different scenes, which will be included in our revised version.

---

### Official Review · Reviewer_46C6 · 2023-07-06

**Soundness:** 4 excellent
**Presentation:** 4 excellent
**Contribution:** 4 excellent
**Rating:** 6
**Confidence:** 4

**Summary:**

This paper proposed a collaborative RGB-D neural implicit SLAM method named CP-SLAM. CP-SLAM is based on neural point representation similar to PointNeRF, which is natually suitable for collaborative SLAM and loop closing. Based on this neural point representation, CP-SLAM is able to perform full SLAM with a single or multiple agents: tracking, mapping, loop detection, loop closing, submap merging, etc. Experiments show that CP-SLAM achieves SOTA results compared with previous collaborative SLAM and neural implicit SLAM methods.

**Strengths:**

1. Using neural point representation in the context of neural implicit SLAM is novel and the motivation of using it is also clear, i.e. more suitable for loop closing and sub-map merging.

2. A full SLAM system that is capable of loop closing, Pose Graph Optimisation and sub-map merging. This is the first time I saw a full SLAM system in the context of neural implicit SLAM.

3. The system seems well engineered, putting many components together and works effectively well.

4. Competitive tracking and mapping results compared with previous collaborative and neural SLAM method.

**Weaknesses:**

The motivation is clear and the method is very well engineered, but from novelty perspective, all the components in CP-SLAM are already there with some incremental modification: e.g. the scene representation is from PointNeRF, Loop detection is from an off-the-shelf NetVLAD model, etc.

**Questions:**

1. From Sec. 3.1 it seems that at each frame a new set of neural points are created densely via back-projection, and there doesn't seem to be any point merging scheme. Then how is CP-SLAM able to achieve 0.62MB parameters size in Tab. 4?

2. Also from Sec. 3.1 it seems that the rendering loss will only optimize the neural features stored in the points but will never affect the locations of the points, which means once created the location of a point will never change unless a loop is detected and a Pose Graph Optimisation is performed. If this is the case, then the entire mapping process would be equivalent to simply back-projecting the depth images?

3. How did you convert your point-based map to mesh for evaluation in Tab. 3?

4. Following my question in 2: if CP-SLAM's map points are obtained explicitly from depth measurement, then it is supposed to have better accuracy than NICE-SLAM and Vox-Fusion (the same reason that TSDF-Fusion has better accuracy than neural SLAM methods). Thus, better accuracy numbers in Tab. 3 is not enough to prove CP-SLAM have "more powerful geometric reconstruction capability". Also it's not clear why the authors didn't report completion.

After rebuttal: the authors have addressed all my questions and concerns.

**Limitations:**

Yes

---

> ### Author Rebuttal · Authors · 2023-08-09
>
> Thanks for your constructive comments and recognition of our strengths. Our responses to your concerns are as follows:
>
> **1. Some components in CP-SLAM are incrementally modified based on existing works. (W1)**
>
> We aim to build a complete collaborative NERF-based SLAM system, focusing more on system-level development. Such work is missing in the field of NERF-based SLAM. We have introduced technical enhancements to both the neural field representation and the system-level pipeline, such as innovatively proposing a keyframe-centric neural point field and a novel learning framework to maintain global consistency. Although the technical novelties are relatively limited, we have made significant contributions to the pipeline as **Reviewer Zpdr** pointed out.
>
> **2. Is there a point cloud fusion strategy in CP-SLAM to reduce memory usage? (Q1)**
>
> In Line 163, we briefly mentioned our point cloud filtering strategy, namely the sparsification strategy, and elaborated on it in **Supp.D**. Specifically, we adopt a grid-based filtering strategy, that is, we perform non-maximum suppression based on the distance from a neural point to the center of a $\rho^3$ cube.
>
> **3. Is the entire mapping process in CP-SLAM equivalent to simply back-projecting the depth images? (Q2)**
>
> CP-SLAM is completely different from traditional visual SLAM methods. As a purely NERF-based SLAM system, 3D points serve solely as spatial carriers of neural feature embeddings for both photometric and geometric information. In CP-SLAM, the mapping process refers to the process of training neural features to better represent the appearance and geometry of the scene and improve the rendered RGB and depth images, rather than back-projecting the depth image. In other words, although the points' location does not change, the encoded scene geometry is changed and improved with the optimized neural features. We dynamically illustrate the above mapping process in our attached supplementary video.
>
> **4. How did you convert your point-based map to mesh for evaluation in Tab.3? (Q3)**
>
> In our implementation, we invoked the built-in **'TSDF-Fusion'** function in Open3D library, supplying it with estimated poses and depth maps from neural volume rendering to generate the final mesh.
>
> **5. If 3D points in CP-SLAM are obtained from depth measurements,  the accuracy improvement in Tab. 3 is not enough. Why the completion is not reported in reconstruction? (Q4)**
>
> Although we utilize ground truth depth maps to obtain neural point locations, as mentioned above, 3D points serve solely as spatial carriers of neural feature embeddings in CP-SLAM. What we input into **`TSDF-Fusion'** is the rendered depth maps from the optimized neural field, not the measurements. In this setting, we achieved better mapping performance than NICE-SLAM and Vox-Fusion. In Lines 274 and 275, we mentioned the reason why we did not evaluate the completion metric. In our loop datasets, the scenes are not completely scanned along with our trajectories, which leads to holes in mesh reconstruction and makes this metric less objective. Therefore, we use 'Depth L1' and 'Acc.' as evaluation metrics of the reconstruction.

---

> > ### Comment · Reviewer_46C6 · 2023-08-19
> >
> > I would like to thank the authors for their response which has answered most of my questions. I still think this paper proposed a novel and interesting method, and I will keep my initial positive rating.
> >
> > However, regarding Q2, 3 and 4, I still have some reservations. Through reading the the authors' answers, now I understand that the neual point-based representation used in this paper is more like a keyframe-centric representation which relies on fusing rendered depth to obtain the final map. While this representation has advantages over feature grids and MLP, it by definition loses the hole-filling ability in unobserved resgions that are outside of cameras field-of-view. But one big advantage of previous neural-SLAM methods like iMAP and NICE-SLAM over traditional methods is they can achieve certain level of hole-filling ability even in the regions outside of cameras' field-of-view, like the hole in ScanNet scene scene0000_00 and several Replica scenes. If this is the case, then the authors need to acknowledge this as an limitation compared to previous methods.
> >
> > Moreover, this also explains why completion is not a good metric here. But it still cannot justify the removal of that metric completely from the table. The authors could simply follow what has been done in Neural-RGBD [1] and GO-Surf [2] by culling out regions that are ouside of any camera frustums before evaluation. In more recent works, ESLAM [3] applied the same culling strategy and Co-SLAM [4] performed more detailed discussion about different strategies used in all previous methods.
> >
> > As such, I think including a detailed discussion and comparision about using neural point-based representation vs. previous neural SLAM methods should definitely be helpful.
> >
> > [1] Neural RGB-D Surface Reconstruction, CVPR 2022
> > [2] GO-Surf: Neural Feature Grid Optimization for Fast, High-Fidelity RGB-D Surface Reconstruction, 3DV 2022
> > [3] ESLAM: Efficient Dense SLAM System Based on Hybrid Representation of Signed Distance Fields, CVPR 2023
> > [4] Co-SLAM: Joint Coordinate and Sparse Parametric Encodings for Neural Real-Time SLAM, CVPR 2023

---

> > > ### Author Response · Authors · 2023-08-20
> > > **Response to Reviewer 46C6 - Hole-filling Ability and Completion Metrics**
> > >
> > > Dear Reviewer 46C6:
> > >
> > > Thank you for your constructive review and positive comments on our work. Here are our responses to your latest comments.
> > >
> > > **1. The hole-filling ability of the neural point-based representation.**
> > >
> > > We agree that our neural point-based representation has weaker hole-filling ability in unobserved regions than the existing methods like NICE-SLAM (hierarchical regular feature grids) and iMAP (coordinate encoding-based ). This arises from the fact that neural points are distributed around the surfaces of observed objects, encoding surrounding scene information within a fixed-radius sphere, which covers smaller regions than iMAP or the feature grid used in NICE-SLAM. We will discuss this limitation in the revised version.
> > >
> > > **2. Completion metric evaluation with the culling strategy.**
> > >
> > > Thank you for your suggestion. As suggested, we have introduced the **'culling strategy (ESLAM)'** into the completion metric evaluation, and the results are listed in the following tables.
> > >
> > > As shown, our method achieves state-of-the-art performance in terms of completion benefiting from high accuracy, while performing on par with the SOTA method (Vox-Fusion) in terms of completion ratio (refer to above 1.), which validates the effectiveness of our method for the single-agent SLAM.
> > >
> > > |   Method   |Office0-loop | Office3-loop | Room0-loop | Room1-loop |
> > > | :--------: | :----------: | :----------: | :--------: | :--------: |
> > > | NICE-SLAM  |1.69     |     2.22     |    1.74    |    1.73    |
> > > | Vox-Fusion |     1.11     |     1.51     |    1.32    |    1.06    |
> > > |    Ours    |**1.04**   |   **1.47**   |  **1.21**  |  **1.01**  |
> > >
> > > **Table. A: Completion [cm] ($\downarrow$) Metric.** The culling strategy is adopted in the completion evaluation. We can observe that our method achieves SOTA performance.
> > >
> > > |   Method   |Office0-loop | Office3-loop | Room0-loop | Room1-loop |
> > > | :--------: |:----------: | :----------: | :--------: | :--------: |
> > > | NICE-SLAM  | 97.22     |    94.82     |   98.14    |   97.98    |
> > > | Vox-Fusion |**99.69**   |  **98.87**   | **99.35**  | **99.84**  |
> > > |    Ours    |99.45     |    98.34     |   99.185   |   99.70    |
> > >
> > > **Table. B: Completion Ratio [<5cm, %] ($\uparrow$) Metric.** The culling strategy is adopted in the completion ratio evaluation. We can observe that our method performs better than NICE-SLAM and on par with Vox-Fusion.

---

> > > > ### Comment · Reviewer_46C6 · 2023-08-20
> > > >
> > > > I would like to thank the authors again for their response to my comments. All my questions have been resolved now, and I will raise my rating to Weak Accept.

---

### Official Review · Reviewer_zwcP · 2023-07-09

**Soundness:** 3 good
**Presentation:** 3 good
**Contribution:** 3 good
**Rating:** 6
**Confidence:** 3

**Summary:**

The authors propose a novel collaborative RGB-D SLAM pipeline, where the map is represented by an implicit field based on Point-NERF that are transformed together with their keyframes allowing pose graph optimization. They use a loop closure detection based on learned features (NetVLAD) also for sub-map alignment. They propose a central federated learning based averaging and refinement of the distributed MLPs after submap alignment.

**Strengths:**

The use of point based features nicely connects the generalization and compression of implicit mapping with the ability to do efficient map refinement. The results on tracking and reconstruction show a large improvement compared to existing works.

**Weaknesses:**

The explanations are sometimes a bit vague. E.g. line 101 what does it mean to incrementally project a pixel to a 3d location?
The initialization of system is not described well.
While the authors describe the averaging and finetuning of the MLP weights, but do not discuss if and how the point features are shared between agents.
Evaluation was only done on synthetic data.

**Questions:**

Why is there another local finetuning and weight averaging done after the central averaging and finetuning?
How is the finetuning done after pose graph optimization? Is it the same as for mapping e.g. eq 9?
Would it not be possible to run ORB-SLAM3 (https://arxiv.org/pdf/2007.11898.pdf) in a multi-map setting using depth from RGB-D as a baseline?
Why not evaluate on real data?
How is the TSDF fusion done to generate the final mesh?
What is the average baseline between frames in the dataset?
How long are the trajectories?

After rebuttal: The authors addressed all of my questions. Thank you.

**Limitations:**

One of the main limitations of framerate/view point change is addressed by the authors.. an initial experiment to showcase the effect would have been nice though.

---

> ### Author Rebuttal · Authors · 2023-08-09
>
> Thank you for your valuable comments. Here are our responses to your concerns.
>
> **1. What does it mean to incrementally project a pixel to a 3d location in line 101? (W1)**
>
> In the SLAM task, both the map and trajectory are incrementally constructed over time. In Line 101, when a new mapping frame arrives, we will back-project a 2D pixel location $\textbf{p}$ to a 3D point $\textbf{P}$ based on the pose, intrinsics, and the depth map, as shown in the following formula:
> $$
> \begin{equation}
> \label{eq1}
>     P = D*K^{-1}*p
> \end{equation}
> $$
>
> Here, $D, K$ represent the ground truth depth and intrinsics, and $p$ is the homogeneous pixel coordinate.
>
> **2. The initialization of the system is not described well. (W2)**
>
> The initialization in NERF-based SLAM refers to utilizing the first frame to construct an initial neural field, similar to the mapping process, but with a few more optimization steps. Specifically, in CP-SLAM, we perform the initialization with 3000$\sim$5000 steps. This practice can stabilize subsequent tracking and mapping with the help of a high-quality initial neural field.
>
>
>
> **3. Authors did not discuss if and how the point features are shared between agents. (W3)**
>
> In our implementation, we built up a real-time updating feature list corresponding to 3D points one by one, and synchronize it to each agent during data exchange.
>
>
>
> **4. Can CP-SLAM be generalized to real-world scenes? (W4+Q4)**
>
> We have experimented on the TUM-RGBD real-world dataset, and the results are shown in the **Tab.G3** in the global response. Following recently proposed NERF-based SLAM works, we chose three scenes: fr1-desk, fr2-xyz, and fr3-office (with loop closure) in TUM-RGBD. The results in **Tab.G3** illustrate that CP-SLAM has also achieved state-of-the-art performance in the real-world setting, and the loop detection and pose graph optimization are equally effective for the real-world scene. It is worth noting that we compared the most recently proposed SOTA methods Co-SLAM [Wang et al. 2023] and ESLAM [Johari et al. 2023] in the TUM-RGBD experiment, and ESLAM encountered an out-of-memory (OOM) issue in the fr2-xyz.
>
>
>
> **5. Why is the local finetuning and weight averaging done after the central averaging and finetuning? (Q1)**
>
> Once loop closure is detected, we will conduct central averaging and finetuning to unify different models into a common domain. Then the finetuned model will be sent to each agent for continuous local tracking and mapping. Once the new keyframe arrives, we will conduct the local update/finetuning given the new observation and the finetuned local model will be sent to the center for central averaging as a new global model for subsequent tracking. This approach only requires lightweight data exchange which further improves real-time performance.
>
>
>
> **6. How is the finetuning done after pose graph optimization? Is it same as Eq. 9? (Q2)**
>
> Finetuning after pose graph optimization is taken as a larger-scale joint mapping. Specifically, we will select a certain number of keyframes from the keyframe pool, which contains keyframes from all agents, for a few-step mapping until the entire pool is traversed. It uses the same principle as **Eq.9**.
>
>
>
> **7. Would it not be possible to run ORB-SLAM3 as a baseline? (Q3)**
>
> As far as we know, ORB-SLAM3 [Campos et al. 2021] is not a collaborative SLAM system. It lacks the capability to process multiple sequences at the same time, thus unable to overcome the efficiency bottleneck in scene exploration. As suggested, we tested ORB-SLAM3 with **‘Atlas'** mode, and we find that only when ORB-SLAM3 fails, it works by saving the disconnected map constructed prior to failure in the atlas as a candidate sub-map, and then creates a new sub-map for subsequent mapping and tracking. If a loop is detected between this new sub-map and one in the atlas, ORB-SLAM3 will perform sub-map fusion.
> For a comprehensive comparison, we have sequentially connected multiple parts from a single multi-agent dataset. We reported multi-agent and single-agent tracking results in **Tab.G1** and **Tab.G2** in the global response. It is clear that CP-SLAM still achieves the best performance in most scenes.
>
> **8. How is the TSDF fusion done to generate the final mesh? (Q5)**
>
> In our implementation, we invoked the built-in **`TSDF-Fusion'** function in Open3D library, supplying it with estimated poses and rendering depth to generate the final mesh.
>
> **9. Average baseslines and the length of trajectories in all datasets. (Q6+Q7)**
>
> Average baselines and the length of trajectories in all datasets are shown in the table below:
>
> |                     | Room-0-loop | Room-1-loop | Office-0-loop | Office-3-loop |
> | :-----------------: | :---------: | :---------: | :-----------: | :-----------: |
> |     Length[$m$]     |    7.69     |    7.82     |     7.69      |     12.95     |
> | Baseline(mean)[$m$] |   0.0039    |    0.004    |    0.0039     |    0.0066     |
>
> **Table A.  Average baselines and the length of trajectories in single-agent datasets.**
>
> |                     |  Part  | Apartment-1 | Apartment-2 | Apartment-0 | Office-0-C |
> | :-----------------: | :----: | :---------: | :---------: | :---------: | :--------: |
> |     Length[$m$]     | Part 1 |    18.04    |    12.12    |    8.71     |    5.66    |
> |                     | Part 2 |    18.61    |    11.24    |    13.34    |    5.66    |
> | Baseline(mean)[$m$] | Part 1 |   0.0046    |   0.0048    |   0.0053    |   0.0037   |
> |                     | Part 2 |   0.0063    |   0.0040    |   0.0035    |   0.0037   |
>
> **Table B. Average baselines and the length of trajectories in two-agent datasets.**

---

### Official Review · Reviewer_q5H1 · 2023-07-10

**Soundness:** 3 good
**Presentation:** 3 good
**Contribution:** 3 good
**Rating:** 6
**Confidence:** 3

**Summary:**

The paper introduces a neural implicit representation based collaborative SLAM technique. They introduce a learnable feature encoding for each point in the 3d scene based on neural rendering subsequently using it for depth and color map estimation. They introduce a centralized learning strategy to share MLPs and learn better weights. They also introduce a LM algorithm based global pose optimization and key frame centric point cloud adjustment.

**Strengths:**

The federated learning strategy improves the neural rendering performance. Map refinement based on keyframe centric neural point field.
By encoding scene geometry and appearance with 3D points, when camera poses are optimized, these features are easily adjusted as well.

**Weaknesses:**

1. The paper only evaluates on one synthetic datasets. It would be nice to see evaluations on more datasets
2. Better and more robust neural rendering techniques could have been used to improve the radiance, depth and color rendering

**Questions:**

1. The paper is not clear whether it includes neural field generation time in tracking/mapping time while comparing with Nice SLAM.
2. The paper lacks citations. For example, in the mapping and tracking section, a similar technique is used in Nice SLAM and I see no mention of it.

**Limitations:**

1.The paper does produce SOTA results on some of the scenes of the dataset chosen. No explanation is given to why it doesn't produce better results on the rest.
2. The paper is not novel enough. It rather combines the best of many other works and produces good results.

---

> ### Author Rebuttal · Authors · 2023-08-09
>
> Thank you for your valuable comments! Here are our responses to your concern.
>
> **1. It would be nice to see evaluations on more datasets. (W1)**
>
> We extended experiments on the TUM-RGBD real-world dataset, and the results are shown in the **Tab.G3** in global response. Following recently proposed NERF-based SLAM works, we chose three scenes: fr1-desk, fr2-xyz, and fr3-office (with loop closure) in TUM-RGBD. The results in **Tab.G3** illustrate that CP-SLAM has also achieved state-of-the-art performance in the real-world setting, and the loop detection and pose graph optimization are equally effective for the real-world scene. It is worth noting that we compared the most recently proposed SOTA methods Co-SLAM [Wang et al. 2023] and ESLAM [Johari et al. 2023] in the TUM-RGBD experiment, and ESLAM encountered an out-of-memory (OOM) issue in the fr2-xyz.
>
> **2. More advanced neural rendering techniques can be used to improve the rendering quality. (W2)**
>
> Several innovative and robust neural rendering techniques have been proposed subsequent to our submission. Nevertheless, given its nature as a collaborative NERF-based SLAM system, we focus more on the overall performance of tracking, mapping, and collaborative cooperation than rendering quality. However, it is interesting to explore more robust rendering techniques in the future.
>
> **3. It is not clear whether time evaluation includes neural field generation time. (Q1)**
>
> Following previous works, such as NICE-SLAM and Vox-Fusion, we evaluated the efficiency of mapping and tracking in Tab.4. All results in Tab.4 solely measure single-frame optimization time in tracking and mapping, excluding the neural field generation time (neural point back-projection and initial neural features inference from CNN).
>
> **4. Some citations are missing in the paper. (Q2)**
>
> In Section 3.1, we described volume rendering, depth loss, and color loss used in tracking and mapping. We presumed these concepts as common sense reached in the NERF-based SLAM field, so we omitted relevant citations. Thanks for your comment, we have checked and added them in Section 3.1.
>
> **5. Novelty is limited. (L1)**
>
> We aim to build a complete collaborative NERF-based SLAM system, focusing more on system-level development. Such work is missing in the field of NERF-based SLAM. we have introduced technical enhancements to both the neural field representation and the system-level pipeline, such as innovatively proposing a keyframe-centric neural point field and a novel learning framework to maintain global consistency. Although the technical novelties are relatively limited, we have made significant contributions to the pipeline as **Reviewer Zpdr** pointed out.

---

> > ### Comment · Reviewer_q5H1 · 2023-08-19
> > **More evaluations and limited novelty**
> >
> > The authors have included more datasets and experiments which achieve SOTA performance. The authors have also corrected the issues with paper writing. Though the authors have developed and achieved a good pipeline for the whole collaborative NERF-based SLAM system, I think the novelty is still a bit limited.

---

> > > ### Author Response · Authors · 2023-08-20
> > > **Further Response to Reviewer q5H1**
> > >
> > > Dear Reviewer q5H1:
> > >
> > > Thank you for your further feedback and for acknowledging our contribution to developing a good pipeline for the collaborative NeRF-based SLAM system.
> > >
> > > Our work may lean more towards system-level technical innovation than theoretical novelties, while we believe it offers a valuable contribution to the community by extending NeRF-SLAM into collaborative scenarios with novel techniques including keyframe-centric neural point representation, distributed-to-centralized learning framework, and flexible fusion strategy among neural sub-maps.

---

> > > > ### Comment · Reviewer_q5H1 · 2023-08-22
> > > > **Raised score to 6.**
> > > >
> > > > In light of the authors' rebuttal, I have raised my score to 6.

---

### Official Review · Reviewer_1EKn · 2023-07-25

**Soundness:** 3 good
**Presentation:** 3 good
**Contribution:** 3 good
**Rating:** 6
**Confidence:** 4

**Summary:**

The paper presents CP-SLAM, a collaborative neural implicit SLAM system developed by the authors. CP-SLAM incorporates neural point-based odometry, loop detection, sub-map fusion, and global refinement. Additionally, the authors propose a novel neural point 3D scene representation with keyframes to enhance map fusion and adjustment. They introduce innovative learning and optimization frameworks to ensure accurate and consistent 3D mapping for cooperative localization and mapping. The performance of CP-SLAM is evaluated on various indoor RGB-D sequences, showcasing its superior mapping and camera tracking capabilities. The contributions made by the authors significantly advance the field of SLAM and offer state-of-the-art solutions.

**Strengths:**

1. CP-SLAM introduces a novel neural point-based representation, enabling dense collaborative neural SLAM.
2. CP-SLAM retains the front-end and back-end modules like traditional SLAM systems, providing a comprehensive pipeline.
3. CP-SLAM outperforms state-of-the-art methods in both localization and reconstruction. The CP-SLAM system supports both single-agent and multi-agent modes, allowing for versatile evaluation.
4. The system is evaluated using datasets based on the Replica scenes, providing a realistic and representative testing environment.
CP-SLAM is compared against recent neural implicit RGB-D SLAM methods in single-agent experiments, showcasing its performance in loop closure.
5. In the two-agent experiments, CP-SLAM is compared with a traditional method, highlighting its collaborative capabilities.
6. The ablation studies are conducted to demonstrate the importance of different modules in the proposed system.
7. The implementation details, such as GPU specifications and parameter settings, are provided, ensuring reproducibility.

**Weaknesses:**

1. CP-SLAM requires significant GPU resources to handle multiple image sequences. The relative pose computation in loop closure relies on existing rendering-based optimization, which may be inaccurate for large viewpoint changes and lead to map fusion drifting.
2. The comparison with traditional methods in the two-agent experiments may not fully represent the performance of other collaborative neural SLAM approaches.
3. CP-SLAM's performance is limited by its GPU resource demand, making it less suitable for resource-constrained devices or environments.
4. Currently, a large-scale experiment is not conducted, but for multi-agents, this is an important point to consider.


**Questions:**

1. How does CP-SLAM handle scalability in terms of the size and complexity of the datasets?
2. Can CP-SLAM be applied to real-world datasets to evaluate its performance in practical scenarios?
3. Are there any specific limitations or challenges in the implementation of CP-SLAM that need to be addressed?
4. Can Nicer-SLAM and DROID-SLAM be compared as a contrast?
5. The article mentioned "Naturally, it is expected that the neural point cloud layout should be rearranged following global pose graph optimization. However, a world-centric point cloud map obviously cannot allow such adjustment. To tackle this limitation, we propose a keyframe-centric neural point field, where each 3D point is associated with a keyframe". Can the authors better explain why a world-centric point cloud map obviously cannot allow such adjustment?

Sincerely,


**Limitations:**

The limitations have been discussed in the conclusion section.

---

> ### Author Rebuttal · Authors · 2023-08-09
>
> Thank you for your careful reading and constructive comments on our paper, and noting the novelty of our CP-SLAM system and its superiority. Here are our responses to your comments.
>
> **1. Large viewpoint change is challenging for NERF-based SLAM methods. (W1)**
>
> As pointed out in Section.5 in our paper, the rendering-based optimization is limited in the face of large viewpoint changes, which remains a bottleneck for other existing works, such as NICE-SLAM and Vox-Fusion. We conduct an ablation study on viewpoint difference in **Fig.4** in the attached **'pdf'** in the global response. It may be partially addressed by the particle filtering algorithm proposed in the latest Loc-NeRF [Maggio et al. ICRA 2023].
>
> **2. More comparisons with traditional methods should be added in the two-agent experiments. (W2)**
>
> Due to the word limit, please refer to **'Comparison with additional traditional methods'** and **Tab.G1** in our global response.
>
> **3. GPU resources are a limitation for CP-SLAM. (W3)**
>
> It is indeed a common weakness in existing neural implicit SLAM works and our CP-SLAM, and we expect further improvements to be made in the future.
>
> **4. A large-scale experiment is not conducted in this paper. (W4)**
>
> In our research, we indeed struggled with limited data resources, and we have not been able to find a public large-scale dataset suitable for collaborative NERF-based SLAM. Multi-agent datasets were mainly captured at large-scale outdoor scenes, such as urban streets. Affected by noisy depth, these available datasets are normally RGB-based, which is not compatible with our RGBD-based CP-SLAM. Moreover, when applied to outdoor scenarios, almost all existing NERF-based SLAM systems encounter pressing issues such as dynamic interference, unbounded scene, and fast motion.  These challenges deserve immediate and critical attention.  It is interesting to work on these limitations and generalize CP-SLAM to outdoor environments, which are out of our current scope.
>
> **5. How does CP-SLAM handle various scene sizes and complexity? (Q1)**
>
> In terms of scene size, in principle, CP-SLAM only needs to incrementally add local neural points to support follow-up tracking. Facing large-scale scenes, our CP-SLAM is more scalable, particularly when compared to the feature grid-based method such as NICE-SLAM. In terms of complexity including extreme lighting and weak texture, NERF-based SLAM exhibits even better robustness over traditional methods, which is evidenced by the failure of traditional methods in Apartment-0 in Tab.1 of our paper.
>
> **6. Can CP-SLAM be generalized to real-world scenes? (Q2)**
>
> Due to the word limit, please refer to **‘Real-world experiment’** and **Tab.G3** in our global response.
>
> **7. Are there any specific limitations or challenges in the implementation of CP-SLAM that need to be addressed? (Q3)**
>
> In collaborative NERF-based SLAM, in addition to the lack of appropriate neural field representation, it is not feasible to simply stack and cascade traditional modules in a plug-and-play manner. Consequently, we have innovatively proposed a keyframe-centric neural point field to address the limitation of sub-map fusion and a novel learning framework to maintain global consistency, etc.
>
> **8. Can Nicer-SLAM and DROID-SLAM be compared as a contrast? (Q4)**
>
> Nicer-SLAM is a non-public and monocular RGB-based neural SLAM system, while our CP-SLAM is a collaborative RGBD-based system. Therefore, we only added Droid-SLAM as a new baseline, and we report experiment results from the official DROID-SLAM in the following table:
>
> |   Method    |    Room-0-loop     |    Room-1-loop     |   Office-0-loop    |   Office-3-loop    |      Average       |
> | :---------: | :----------------: | :----------------: | :----------------: | :----------------: | :----------------: |
> | DROID-SLAM  | **0.28/0.26/0.23** | **0.26/0.22/0.19** | **0.45/0.41/0.37** | **0.33/0.29**/0.30 | **0.33/0.30/0.27** |
> | Ours(w$/$o) |   0.61/0.56/0.54   |   0.51/0.48/0.52   |   0.67/0.63/0.67   |   0.38/0.32/0.27   |   0.54/0.50/0.50   |
> | Ours(w$/$)  |   0.48/0.44/0.43   |   0.44/0.40/0.46   |   0.56/0.53/0.56   | 0.37/0.31/**0.27** |   0.46/0.42/0.43   |
>
> **Table.A Single-agent Tracking Performance compared with DROID-SLAM**
>
> |   Method   |  Room-0-loop  |  Room-1-loop  | Office-0-loop | Office-3-loop |    Average    |
> | :--------: | :-----------: | :-----------: | :-----------: | :-----------: | :-----------: |
> | DROID-SLAM |    58/6.47    |    3/4.17     |    34/8.12    |    61/7.99    |    46/6.69    |
> |    Ours    | **0.32/1.53** | **0.23/1.20** | **0.22/1.21** | **0.76/1.7**  | **0.38/1.41** |
>
> **Table.B Mapping Performance compared with DROID-SLAM**
>
> We observe that CP-SLAM performs slightly worse than DROID-SLAM in terms of tracking, but much better than DROID-SLAM in reconstruction. In addition, DROID-SLAM cannot support multi-agent collaboration.
>
> **9. Why the world-centric model cannot allow neural point cloud adjustment after pose graph optimization? (Q5)**
>
> Keyframe-centric model is a crucial module in the proposed CP-SLAM system. Specifically, the neural point field is composed of two parts: the 3D point cloud and the corresponding feature embeddings which encode the surrounding regions. Following pose graph optimization (PGO), it is necessary to refine previously selected point positions based on more accurate poses, so as to reuse previously optimized feature embeddings. This requires a clear correspondence between 3D points and mapping frames. If a world-centric model is adopted, neural points cannot be refined along with camera poses due to the absence of such correspondences, so it may be possible to recompute all 3D points and learn all neural features from scratch for stable tracking and mapping after PGO.

---

> > ### Comment · Reviewer_1EKn · 2023-08-11
> > **Comment**
> >
> > Dear authors,
> >         thank you for your responses and added analyses. We appreciate that the authors add the comparisons against DROID-SLAM and the comparisons in the global response. These analyses should be added to the final version. We would like to maintain the positive rating and suggest that the authors make the code publicly available to foster future research in collaborative SLAM.
> >
> > Sincerely,

---

> > > ### Author Response · Authors · 2023-08-11
> > > **Thank you!**
> > >
> > > Dear Reviewer 1EKn,
> > >
> > > Thank you for your constructive suggestions and positive evaluation of our work again. As suggested, we will release our code upon acceptance and integrate these new comparisons into the revised version.

---

### Official Review · Reviewer_Zpdr · 2023-07-28

**Soundness:** 3 good
**Presentation:** 3 good
**Contribution:** 3 good
**Rating:** 5
**Confidence:** 4

**Summary:**

There have been some papers on getting dense Visual SLAM with neural fields published in the last couple of years. In this work, the authors proposed a set of additions to prior pipelines that get the NN SLAM closer to the traditional approaches concerning the overall framework. Specifically, the authors addressed loop detection, pose graph optimization, bundle adjustment, and collaborative SLAM.

**Strengths:**

1) Visual-SLAM with neural fields has the huge benefit of providing a dense map of the environment. The topic is relevant to computer vision, robotics, and I believe it fits the NeurIPS scope.
2) The paper is well-written, easy to follow, and well-organized. Related work is OK.
3) Although the technical novelties are relatively limited, in my opinion, the contributions to the pipeline are significant.

**Weaknesses:**

1) As written above, the technical contributions are limited. The paper focuses more on adding new steps to prior Visual SLAM pipelines.
2) The inclusion of the NetVLAD (not new) as a tool for loop detection looks like a trick. It is not elegant, in the sense that it is a parallel feature extractor that (maybe) could be done using the neural field encoders.
3) Only one dataset and 4 sequences are used, which is small concerning quantity and variety.
4) I would prefer to see more comparisons with previous traditional methods (without neural fields).
5) Figures 5 and 6 are not easily readable.

**Questions:**

1) I would like the authors to explain what is the motivation for the inclusion of NetVLAD. Can't those descriptors be obtained from the neural field?
2) It is written that a pre-trained NetVLAD was used. It would be important to give more details about this training and how it generalizes to different scenes.
3) Usually, with classical V-SLAM, the bundle adjustment estimates both the pose of the cameras and the 3D map simultaneously. It seems that the authors are doing this alternatively. I would like to see some comments on this.
4) Why are the results for Apartment 2 so bad for the proposed approach? I would like the authors to comment on this.
5) It seems that traditional V-SLAM methods using RGB-D are missing (such as OrbSLAM). I would like to see some comments on this.
6) Maybe I missed something. But it is not clear to me how is the error in Tab. 2 computed.

**Limitations:**

Limitations are adequately addressed.

---

> ### Author Rebuttal · Authors · 2023-08-09
>
> Thank you for your valuable comments and for appreciating our contributions and writing. Here are our responses to your comments.
> **1. The paper focuses more on adding new steps to prior Visual SLAM pipelines. (W1)**
> We aim to build a complete collaborative NERF-based SLAM system, focusing more on system-level development. Such work is missing in the field of NERF-based SLAM. CP-SLAM is the first collaborative neural implicit SLAM system, consisting of neural point-based odometry, loop detection, sub-map fusion, and global pose graph optimization. In collaborative NERF-based SLAM, in addition to the lack of appropriate neural field representation, it is not feasible to simply stack and cascade traditional modules in a plug-and-play manner. Consequently, we have introduced technical enhancements to both the neural field representation and the pipeline of CP-SLAM, such as a novel keyframe-centric neural point field and a novel learning framework to maintain global consistency.
>
> **2. Why not use the neural field encoders, instead of NetVLAD, for loop detection? (W2+Q1)**
>
> We have simply tried to take neural point features as descriptors for loop detection. However, neural features only encode local surrounding information, rather than a global descriptor like NetVLAD. This leads to poor distinctiveness of these neural features and further results in false positive matches which makes the loop detection unstable.
>
> **3. Dataset is limited. (W3)**
> We have experimented on the TUM-RGBD real-world dataset, and the results are shown in **Tab.G3** in the global response. Following recently proposed NERF-based SLAM works, we chose three scenes: fr1-desk, fr2-xyz, and fr3-office (with loop closure) in TUM-RGBD. The results in **Tab.G3** illustrate that CP-SLAM has also achieved state-of-the-art performance in the real-world setting, and the loop detection and pose graph optimization are equally effective for the real-world scene. It is worth noting that we compared the most recently proposed SOTA methods Co-SLAM [Wang et al. 2023] and ESLAM [Johari et al. 2023] in the TUM-RGBD experiment, and ESLAM encountered an out-of-memory (OOM) issue in the fr2-xyz.
>
> **4. More comparisons with traditional methods should be added. (W4+Q5)**
> We have compared with two more baselines including ORB-SLAM3 [Campos et al. 2021]  and Swarm-SLAM [Lajoie et al. 2023] for the two-agent experiment to demonstrate our performance and the corresponding results are shown in **Tab.G1** in the global response. It is clear that our method still performs the best. To be noted, ORB-SLAM3 is not a collaborative SLAM system. It lacks the capability to process multiple sequences at the same time, thus unable to overcome the efficiency bottleneck in scene exploration. We evaluate ORB-SLAM3 just for a more comprehensive comparison.
>
> **5. Figures 5 and 6 are not easily readable. (W5)**
> Thank you for your comment, we have improved these two figures for a better visual experience, please refer to **Fig.1** and **Fig.2** in the attached **`pdf'** in our global response.
>
> **6. Training details and generalization of NetVLAD. (Q2)**
> The pre-trained NetVLAD model used in our paper comes from the open-source project proposed by Paul et al. In our test, we observed its robustness across various scenes and insensitivity to hyperparameters.  Actually, the pre-trained NetVLAD was also normally adopted for loop detection in other SLAM systems like DOOR-SLAM [Lajoie et al. 2020] and DSLAM [Cieslewski et al. 2018].
>
> **7. Separate pose and map optimization. (Q3)**
> During mapping, we select multiple frames for joint optimization based on the covisible correlations. Considering the limitation of GPU memory, the total number of rays is fixed, that is, only a few rays can be allocated to each frame. In this case, considering the irregular distribution of neural points compared to the regular feature grid, the pose adjustment is prone to the sub-optimal solution if it is done together with mapping, so for more stable pose optimization, we choose to perform tracking separately with more rays after mapping.
>
> **8. Pose drift in Apartment-2. (Q4)**
> Existing NERF-based SLAM methods often construct an L1 loss between rendered depth and the ground truth depth, relying on local depth variance to guide pose estimation towards a correct solution. However, in the **`Apartment2-part2'** sequence, the camera frequently moves parallel to walls with uniform depth, aggravating non-convexity in the depth loss and resulting in ambiguous solutions. This is a common problem in recently proposed NERF-based SLAM approaches, and we hope to overcome this bottleneck in future work.
>
> **9. Error calculation method in Tab.2. (Q6)**
> The results in Tab.2 were obtained in an origin-aligned manner with the **EVO** toolbox for proper drift/loop closure evaluation. Specifically, after aligning the origin poses, we calculate the absolute trajectory error (ATE) for each pose and compute the RMSE/Mean/Median values.

---

> ### Comment · Area_Chair_km3S · 2023-08-20
>
> Dear Reviewer Zpdr,
>
> Many thanks for your review on the paper. Your rating is borderline kind. Could please read the author's rebuttal and give your opinion and final rating?
>
> Thank you very much,
>
> AC

---

### Author Rebuttal · Authors · 2023-08-09

We thank all the reviewers for their valuable comments and appreciate the positive comments like “the contributions to the pipeline are significant” **(Reviewer Zpdr)**, "CP-SLAM introduces a novel neural point-based representation, enabling dense collaborative neural SLAM." **(Reviewer 1EKn)**, "The federated learning strategy improves the neural rendering performance. " **(Reviewer q5H1)**, "The use of point based features nice connects implicit mapping with the ability to do efficient map refinement" **(Reviewer zwcP)**, "This is the first time I saw a full SLAM system in the context of neural implicit SLAM." **(Reviewer 46C6)**, "Improved Performance in both camera tracking and mapping." **(Reviewer jr2H)**.  In this global response, we address the common concerns regarding our experiment and will separately address specific comments raised.

- **Comparison with additional traditional methods:** In the multi-agent experiment, we added two traditional baselines including ORB-SLAM3 [Campos et al. 2021]  and Swarm-SLAM [Lajoie et al. 2023] as suggested for a more comprehensive comparison. For the single-agent experiment, we compared with ORB-SLAM3. The results are listed in **Table G1** and **G2**. We can observe that our method still performs the best compared to these methods. It is worth noting that ORB-SLAM3 is not a collaborative SLAM system. It lacks the capability to process multiple sequences at the same time, thus unable to overcome the efficiency bottleneck in scene exploration.
- **Real-world experiment:** We have experimented on the TUM-RGBD real-world dataset. From **Table G3**, we can see that our CP-SLAM has also achieved state-of-the-art performance in the real-world setting, and the loop detection and pose graph optimization are effective for the real-world scene. The most recently proposed SOTA methods, i.e., Co-SLAM [Wang et al. 2023] and ESLAM [Johari et al. 2023], are evaluated for comparison in the TUM-RGBD experiment.

|   Method   |    Part     |    Apartment-1     |    Apartment-2     |    Apartment-0     | Office-0-C(SingleRoom) |
| :--------: | :---------: | :----------------: | :----------------: | :----------------: | :--------------------: |
|  CCM-SLAM  |  **Part1**  |   2.12/1.94/1.74   | **0.51/0.45/0.40** |       -/-/-        |     9.84/8.23/6.41     |
| ORB-SLAM3  |             |   4.93/4.65/5.01   |   1.35/1.05/0.65   |   0.67/0.58/0.47   |     0.66/0.62/0.62     |
| Swarm-SLAM |             |   4.62/4.17/3.90   |   2.69/2.48/2.34   |   1.61/1.33/1.09   |     1.07/0.96/0.98     |
| Ours(w/o)  |             |   1.15/0.99/0.88   |   1.45/1.34/1.36   | 0.70/0.48/**0.27** |     0.71/0.62/0.67     |
|  Ours(w/)  |             | **1.11/0.95/0.81** |   1.41/1.30/1.36   | **0.62/0.47**/0.30 |   **0.50/0.46/0.55**   |
|  CCM-SLAM  |  **Part2**  |   9.31/6.36/5.57   | **0.48/0.43/0.38** |       -/-/-        |   0.76/**0.36/0.16**   |
| ORB-SLAM3  |             |   4.93/4.04/3.80   |   1.36/1.24/1.11   |   1.46/1.11/0.79   |   **0.54**/0.49/0.47   |
| Swarm-SLAM |             |   6.50/5.27/4.39   |   8.53/7.59/7.10   |   1.98/1.48/0.94   |     1.76/1.55/1.83     |
| Ours(w/o)  |             |   2.12/2.05/2.23   |   2.54/2.45/2.60   |   1.61/1.55/1.70   |     1.02/1.03/0.99     |
|  Ours(w/)  |             | **1.72/1.61/1.46** |   2.41/2.33/2.44   | **1.28/1.17/1.37** |     0.79/0.74/0.70     |
|  CCM-SLAM  | **Average** |   5.71/4.15/3.66   | **0.49/0.44/0.39** |       -/-/-        |     5.30/4.29/3.29     |
| ORB-SLAM3  |             |   4.93/4.35/4.41   |   1.36/1.15/0.88   | 1.07/0.85/**0.63** |   **0.60/0.56/0.55**   |
| Swarm-SLAM |             |   5.56/4.72/4.15   |   5.61/5.04/4.72   |   1.80/1.41/1.02   |     1.42/1.26/1.41     |
| Ours(w/o)  |             |   1.64/1.52/1.56   |   2.00/1.90/1.98   |   1.16/1.02/0.99   |     0.86/0.81/0.83     |
|  Ours(w/)  |             | **1.42/1.28/1.14** |   1.91/1.82/1.90   | **0.95/0.82**/0.84 |     0.65/0.60/0.63     |

**Table G1: Two-agent Tracking Performance.  ATE RMSE/Mean/Median [cm] are reported.**

|   Method    |    Room-0-loop     |    Room-1-loop     |   Office-0-loop    |   Office-3-loop    |      Average       |
| :---------: | :----------------: | :----------------: | :----------------: | :----------------: | :----------------: |
|  NICE-SLAM  |   1.27/1.15/1.09   |   1.74/1.61/1.66   |   2.27/1.91/1.82   |   3.19/2.77/2.28   |   2.12/1.86/1.71   |
| Vox-Fusion  |   0.82/0.77/0.78   |   1.35/1.30/1.25   |   0.99/0.94/0.95   |   0.82/0.74/0.73   |   0.99/0.94/0.93   |
|  ORB-SLAM3  |   0.54/0.52/0.53   | **0.21/0.19/0.19** |   0.58/0.51/0.52   |   0.89/0.80/0.84   |   0.56/0.51/0.52   |
| Ours(w$/$o) |   0.61/0.56/0.54   |   0.51/0.48/0.52   |   0.67/0.63/0.67   | 0.38/0.32/**0.27** |   0.54/0.50/0.50   |
| Ours(w$/$)  | **0.48/0.44/0.43** |   0.44/0.40/0.46   | **0.56/0.53/0.56** | **0.37/0.31/0.27** | **0.46/0.42/0.43** |

**Table G2: Single-agent Tracking Performance.  ATE RMSE/Mean/Median [cm]  are reported.**

|  Method   | fr1-desk (w/o loop) | fr2-xyz (w/o loop) | fr3-office (w/ loop) |      Average       |
| :-------: | :-----------------: | :----------------: | :------------------: | :----------------: |
|  Co-SLAM  |   7.10/6.83/6.79    |   4.05/3.76/3.47   |    5.58/5.06/4.57    |   5.58/5.22/4.95   |
|   ESLAM   | **6.81/6.56/6.88**  |   Fail/Fail/Fail   |    4.23/3.91/3.73    |         -          |
| Ours |   7.84/7.34/7.12    | **3.93/3.50/3.29** |     **3.84/3.47/3.39**    | **5.20/4.77/4.60** |

**Table G3: Single-agent Tracking Performance on the TUM-RGBD dataset.** In the real-world experiment, we can find that CP-SLAM still performs the best. Besides, ESLAM fails in the fr2-xyz because of OOM (out of memory). '-' indicates that metrics cannot be evaluated due to ESLAM failures.

---

### Decision · Program_Chairs · 2023-09-21

**Decision:**

Accept (poster)

**Comment:**

The paper presents CP-SLAM, a collaborative implicit neural SLAM system. The authors propose a neural point based 3D scene representation in which each point maintains a learnable neural feature for scene encoding and is associated with a certain keyframe. The authors also propose a distributed-to-centralized learning strategy to improve consistency and cooperation. A  global optimization framework is also proposed to improve the system's accuracy.

The paper received five “Weak accept” ratings and one “Borderline accept” rating.

Reviewer q5H1 gave “Weak Accept”. Reviewer q5H1 thinks the federated learning strategy improves the neural rendering performance. And, Reviewer q5H1 thinks some features are easily adjusted by encoding scene geometry and appearance with 3D points. But also, Reviewer q5H1 wants to see evaluations on more datasets. The authors gave their rebuttals. After reading them, Reviewer q5H1 had no further questions.

Reviewer jr2H gave “Weak Accept”. Reviewer jr2H thinks a neural point cloud representation for scene encoding in the proposed framework is a significant contribution. Reviewer jr2H thinks the proposed system can handle various challenges typically encountered in SLAM tasks. And, Reviewer jr2H thinks the incorporation of a distributed-to-centralized learning strategy is an effective solution for improving consistency and cooperation in collaborative implicit SLAM. But also, Reviewer jr2H wants to see more experiments. The authors gave their rebuttals. After reading them, Reviewer jr2H had no further questions.

Reviewer 46C6 gave “Weak Accept”. Reviewer 46C6 thinks this paper proposes the first full SLAM system in the context of neural implicit SLAM. Reviewer 46C6 thinks the system is well engineered, putting many components together and works effectively well. Reviewer 46C6 thinks the system achieves competitive tracking and mapping results compared with previous collaborative and neural SLAM methods. But also, Reviewer 46C6 thinks the novelty of this paper is not enough. The authors gave their rebuttals. After reading them, Reviewer 46C6 had no further questions.

Reviewer zwcP gave “Weak Accept”. Reviewer zwcP thinks the use of point based features nicely connects the generalization and compression of implicit mapping with the ability to do efficient map refinement. Reviewer zwcP thinks the results on tracking and reconstruction show a large improvement compared to existing works. But also, Reviewer zwcP thinks the authors should further clarify some concepts and should include more experimental results. The authors gave their rebuttals. After reading them, Reviewer zwcP had no further questions.

Reviewer 1EKn gave “Weak Accept”. Reviewer 1EKn thinks this paper introduces a novel neural point-based representation, enabling dense collaborative neural SLAM. Reviewer 1EKn thinks the system achieves state-of-the-art performance. Reviewer 1EKn thinks the implementation details are enough. But also, Reviewer 1EKn thinks the performance of the system is limited by its GPU resource demand. The authors gave their rebuttals. After reading them, Reviewer 1EKn had no further questions.

Reviewer Zpdr gave “Borderline accept”. Reviewer Zpdr thinks the paper is well-written, easy to follow, and well-organized. Reviewer Zpdr thinks the contributions are significant. But also, Reviewer Zpdr thinks the novelty of this paper is limited. Reviewer Zpdr wants to see more experimental results. The authors gave their rebuttals. After reading them, Reviewer Zpdr had no further questions.

In summary, the theoretical novelty is not enough and more contribution is on a system-level development.  Please make revisions and address all other questions raised by the reviewers. Then, the paper can be accepted by NeurIPS.